# On the Effect of Negative Gradient in Group Relative Deep Reinforcement Optimization

**Wenlong Deng**[1,2], **Yi Ren**[1], **Muchen Li**[1]
**Danica J. Sutherland**[1,3], **Xiaoxiao Li**[1,2†], **Christos Thrampoulidis**[1†]

[1]University of British Columbia, [2]Vector Institute, [3]Amii

[†]Corresponding author

## Abstract

Reinforcement learning (RL) has become popular in enhancing the reasoning capabilities of large language models (LLMs), with Group Relative Policy Optimization (GRPO) emerging as a widely used algorithm in recent systems. Despite GRPO's widespread adoption, we identify a previously unrecognized phenomenon we term Lazy Likelihood Displacement (LLD), wherein the likelihood of correct responses marginally increases or even decreases during training. This behavior mirrors a recently discovered misalignment issue in Direct Preference Optimization (DPO), attributed to the influence of negative gradients. We provide a theoretical analysis of GRPO's learning dynamic, identifying the source of LLD as the naive penalization of all tokens in incorrect responses with the same strength. To address this, we develop a method called NTHR, which downweights penalties on tokens contributing to the LLD. Unlike prior DPO-based approaches, NTHR takes advantage of GRPO's group-based structure, using correct responses as anchors to identify influential tokens. Experiments on math reasoning benchmarks demonstrate that NTHR effectively mitigates LLD, yielding consistent performance gains across models ranging from 0.5B to 3B parameters.

## 1 Introduction

Reinforcement learning (RL) has become increasingly popular for improving reasoning capabilities of large language models (LLMs) [4, 9, 23]. In particular, Group Relative Policy Optimization (GRPO) [22] has emerged as a widely adopted algorithm in RL training pipelines for reasoning tasks. Models such as DeepSeek-R1 [4], DeepSeek-Math [22], Med-R1 [11], and Search-R1 [10] have successfully employed GRPO to achieve notable performance gains in code generation, mathematical problem solving, medical reasoning, and retrieval-augmented generation. These successes highlight GRPO's growing importance as a tool for aligning models to task-specific behaviors through rule-based or heuristic rewards.

Despite its empirical success, the optimization behavior of GRPO remains insufficiently studied, in part due to its recency. In this work, we focus on the impact of *negative gradients* in GRPO, which arise when the advantage function is negative—indicating that the selected action is worse than the average action at a given state (e.g., an incorrect or suboptimal response). In such cases, the learning algorithm decreases the probability of selecting this action. We draw a conceptual connection to Direct Preference Optimization (DPO) [18], which introduces negative gradients when penalizing dis-preferred responses. Recent studies [19, 17, 30, 20] have revealed that negative gradients can lead to an arguably counterintuitive behavior: a simultaneous decrease in the probabilities of preferred responses ($y^+$). Motivated by this insight, we examine whether similar gradient dynamics in GRPO might cause analogous issues and investigate their impact on policy learning. This effect,

termed misalignment [20] or likelihood displacement [19], is undesirable as it often degrades model performance by diverting probability mass away from optimal responses. In this work, we empirically observe (see Fig. 1) that GRPO can suffer from what we call Lazy Likelihood Displacement (LLD): a failure to sufficiently increase, or even a decrease in, the likelihood of correct answers during training. To explain this phenomenon, we provide an analysis of GRPO's update dynamics and identify that LLD stems from penalizing certain tokens in dispreferred responses. These updates inadvertently reduce the likelihood of correct answers due to shared structural or semantic features between positive and negative responses.

While several approaches have been proposed to mitigate the reduced probabilities of preferred responses ($y^+$) in DPO [19, 20], they either compromise data efficiency or rely on training procedures that are not directly compatible with GRPO's online learning paradigm. For instance, Razin et al. [19] recommend discarding training examples where the likelihood of positive responses $y^+$ decreases, thereby avoiding harmful updates—but this strategy reduces data utilization and does not address cases where the increase in the likelihood of $y^+$ is insufficient. Similarly, Ren and Sutherland [20] propose modifying DPO into a more online-compatible setup by training on both $y^+$ and $y^-$, which effectively mitigates the "squeezing effect" imposed by off-policy negative gradient. However, GRPO already employs online response sampling, and our empirical analysis reveals that it still suffers from the LLD effect in a subset of training examples (see Fig. 1).

To effectively address LLD in GRPO, we introduce a negative token hidden reward (NTHR) for *selective token penalization*, which reduces the penalty on tokens in incorrect responses that most strongly contribute to lowering the likelihood of correct ones. Interestingly, we find that these influential tokens often align closely with key reasoning steps in the correct responses (see Fig. 3). Our approach NTHR provides a fine-grained and targeted mitigation strategy that mitigates LLD during training. We evaluate GRPO+NTHR through per-sample training and compare it against baseline methods such as random token dropping and standard GRPO across multiple model sizes. We demonstrate that GRPO+NTHR successfully mitigates LLD and leads to improved generalization on held-out mathematical reasoning problems. In summary, our contributions are as follows:

• **GRPO exhibits LLD for correct responses.** We empirically show that GRPO suffers from Lazy Likelihood Displacement (LLD), an extended form of the misalignment phenomenon previously observed only in DPO.

• **Identifying the source of LLD.** We provide a theoretical explanation for LLD in GRPO, identifying its cause as a penalization of shared reasoning or correct tokens in dispreferred responses.

• **NTHR selective token penalization.** We introduce Negative Token Hidden Reward (NTHR) for selective *token* penalization that effectively mitigates LLD without sacrificing data efficiency.

• **Empirical validation on math reasoning tasks.** We demonstrate that GRPO+NTHR consistently outperforms GRPO on math benchmarks across model sizes ranging from 0.5B to 3B.

## 2 Preliminaries and Related Work

### 2.1 GRPO

GRPO loss, introduced in DeepSeek-Math [4] and DeepSeek-R1 [22], enhances fine-tuning by refining how reward and loss are calculated. Concretely, unlike traditional Proximal Policy Optimization (PPO) [21], GRPO eliminates the need for value function estimation, employing group-relative rewards for a more nuanced optimization process.

For a query-answer pair $(\boldsymbol{x}, \boldsymbol{a})$, the policy $\pi_\theta$ samples $G$ responses $\{\boldsymbol{y}_i\}_{i=1}^G$. Each $\boldsymbol{y}_i$ consists of a sequence of $|\boldsymbol{y}_i|$ tokens, and we denote $\boldsymbol{y}_{i,<k}$ the subsequence of the first $k$ tokens. Let $r_i$ denote the reward for response $\boldsymbol{y}_i$. The advantage of the $i$-th response is computed by normalizing the group-level rewards $\{r_i\}_{i=1}^G$ and is the same for each token $k = 1, \ldots, |\boldsymbol{y}_i|$. Concretely, $\hat{A}_{i,k} := \frac{r_i - \mu}{\sigma}$, with $\mu = \widehat{\mathbb{E}}[\{r_i\}_{i=1}^G]$ and $\sigma = \sqrt{\widehat{\text{Var}}[\{r_i\}_{i=1}^G]}$ being the empirical average and standard deviation of

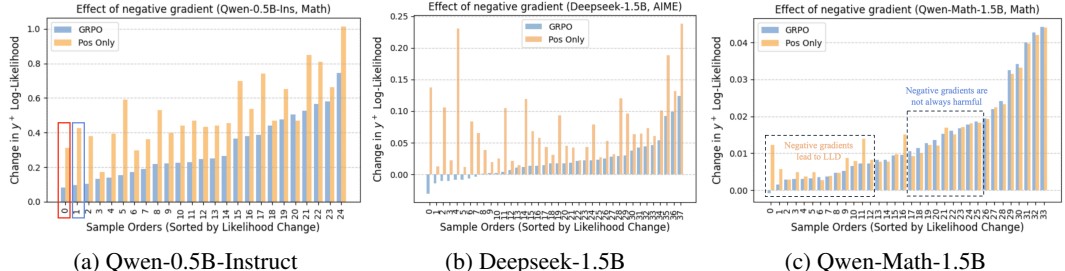

| (a) Qwen-0.5B-Instruct | (b) Deepseek-1.5B | (c) Qwen-Math-1.5B |

Figure 1: We show that negative gradients can lead to small or reduced likelihood change of positive samples in GRPO. The log-likelihood gains achieved by `Pos Only` training (orange) are significantly higher than those from GRPO (blue) for Qwen-0.5B-Ins (a) and Deepseek-1.5B (b). In Qwen-Math-1.5B (c), samples with small or reduced $\Delta(\boldsymbol{x})$ (left) are primarily influenced by negative gradients, as evidenced by their larger $\Delta(\boldsymbol{x})$ in the `Pos Only` setup. However, some samples on the right show smaller $\Delta(\boldsymbol{x})$ than in GRPO, indicating that negative gradients are not always harmful.

the rewards. The GRPO objective $\mathcal{J}_{\text{GRPO}}(\theta)$ is then defined as:

$$\underset{\substack{(\boldsymbol{x},\boldsymbol{a})\sim\mathcal{D} \\ \{\boldsymbol{y}_i\}_{i=1}^{G}\sim\pi_{\theta_{\text{old}}}(\cdot|\boldsymbol{x})}}{\mathbb{E}} \left[ \frac{1}{\sum_{i=1}^{G}|\boldsymbol{y}_i|} \sum_{i=1}^{G}\sum_{k=1}^{|\boldsymbol{y}_i|} \min\left(\gamma_{i,k}(\theta)\hat{A}_{i,k}, \hat{A}_{i,k}\cdot\text{clip}\left(\gamma_{i,k}(\theta), 1-\varepsilon, 1+\varepsilon\right)\right) \right] \quad (1)$$

where $\varepsilon$ is a clipping hyperparameter, $\text{clip}(\cdot)$ is the clipping operation, and $\gamma_{i,k}(\theta) = \frac{\pi_\theta(\boldsymbol{y}_{i,k}|\boldsymbol{x},\boldsymbol{y}_{i,<k})}{\pi_{\theta_{\text{old}}}(\boldsymbol{y}_{i,k}|\boldsymbol{x},\boldsymbol{y}_{i,<k})}$ is the likelihood ratio between the current policy $\pi_\theta$ and the old policy $\pi_{\theta_{\text{old}}}$.

## 2.2 Positive and Negative Gradient

The GRPO loss includes several regularization mechanisms designed to stabilize training, such as the KL term and the clipping operation. However, it is common to treat the log-likelihood component of the generated responses as responsible for "gathering knowledge," while viewing the regularization components as responsible for "stabilizing the training procedure." Without loss of generality, we focus on the online training setting for GRPO, in which $\gamma_{i,t} \approx 1$ and $\min(\cdot, \text{clip}(\cdot))$ can be safely neglected. This simplification is justified by two observations. First, GRPO inherently operates in a near-online fashion, as it generates new samples on-the-fly and typically uses mini-batches that are comparable in scale to the full batch. Second, prior work [1] has shown that omitting the clipping operation does not degrade performance. Additionally, related studies [1, 8] have demonstrated that the KL term can be omitted when other hyperparameters are carefully tuned. With this simplification, we can better understand the interactions between positive and negative responses in one roll-out. The subtle differences between GRPO and its variants, e.g., DAPO [29], Dr.GRPO [13], GPG [1], etc., could also be well interpreted.

Specifically, the GRPO's objective gradient $\nabla_\theta \mathcal{J}_{\text{GRPO}}(\theta)$ with respect to $\theta$ can be approximated as $\nabla_\theta \mathbb{E}\left[\frac{1}{\sum_{i=1}^{G}|\boldsymbol{y}_i|}\sum_{i=1}^{G}\sum_{k=1}^{|\boldsymbol{y}_i|}\hat{A}_{i,k}\gamma_{i,k}(\theta)\right]$ which using $\nabla_\theta\pi_\theta = \pi_\theta\nabla_\theta\log\pi_\theta$ further simplifies to

$$\mathbb{E}\left[\frac{1}{\sum_{i=1}^{G}|\boldsymbol{y}_i|}\sum_{i=1}^{G}\sum_{k=1}^{|\boldsymbol{y}_i|}\underbrace{\hat{A}_{i,k}\gamma_{i,k}(\theta)}_{\text{constant}}\nabla_\theta\log\pi_\theta(\boldsymbol{y}_{i,k}\mid\boldsymbol{x},\boldsymbol{y}_{i,<k})\right].$$

This expression allows us to interpret GRPO using a similar framework as proposed by Ren and Sutherland [20], i.e., imposing a positive or a negative pressure on $\boldsymbol{y}_{i,k}$. Since $\gamma_{i,k}$ is a ratio of two probabilities (and hence it must be positive), the sign of the equivalent learning rate on $\boldsymbol{y}_{i,k}$ is determined by $\hat{A}_{i,k}$. Then, following a common practice that the reward is 1 for correct responses and 0 for incorrect ones, it is safe to conclude that all the tokens in the correct responses impose positive gradients while tokens in the wrong responses impose negative ones.

# 3 The Effect of Negative Gradients

We begin by empirically analyzing the impact of negative gradients in GRPO [4] on the likelihood of generating correct responses. Let each question $x$ be associated with $G$ generated samples, of which $N^+ := N^+(x)$ give the correct answer so we call them *positive* and denote by $y_i^+, i \in [N^+]$, and the rest $N^- L = G - N^+$ are incorrect we call them *negative* and denote by $y_j^-, j \in [N^-]$. Positive/negative samples are given rewards $r_i = 1$ and $r_j = 0$ respectively.

We conduct experiments using math-reasoning tasks to assess how negative gradient in GRPO training influences the log-likelihood of correct responses. Specifically, we examine a range of model-dataset combinations that differ in terms of dataset difficulty and model scale: Qwen-2.5-0.5B [25], Qwen-2.5Math-1.5B [26] with the MATH dataset [7], and Deepseek-1.5B [4] with the AIME dataset. For each question, we generate 8 response rollouts, filtering out samples where all responses are either entirely correct or incorrect. We retain only those examples containing a mix of both.

Our analysis focuses on the GRPO baseline (illustrated by the blue bars in Fig. 4). To precisely characterize the learning dynamics of the algorithm, we reinitialize the model parameters $\theta$ for each individual sample, perform a single GRPO update to obtain the updated parameters $\theta'$, and evaluate the average change in the log-likelihood of the correct responses:

$$\Delta(x) := \frac{1}{N^+} \sum_{i=1}^{N^+} \left[ \ln \pi_{\theta'}(y_i^+ \mid x) - \ln \pi_\theta(y_i^+ \mid x) \right], \tag{2}$$

for an input question $x$ with $N^+$ correct responses $y_i^+, i \in [N^+]$. As illustrated in Fig. 1, some correct responses experience either a decline or only a marginal improvement in log-likelihood after training. For example, in Deepseek-1.5B (Fig. 1b), a significant portion of samples exhibit reduced likelihoods (negative values in the plot). Similarly, in Qwen-Math-1.5B and Qwen-0.5B, many samples show only minor changes in likelihood. A closer analysis shows the following.

**Negative gradients lead to small or reduced likelihood change.** In order to isolate the effect of negative gradients, we introduce a variant which we call `Pos Only`, where negative advantages are masked (i.e., values of $\hat{A}_{i,k} < 0$ are set to 0), effectively removing the influence of negative gradients during training. Fig. 1 shows the `Pos Only` variant in orange bars. The log-likelihood gains under `Pos Only` training are substantially higher than those from GRPO—particularly on the left side of the plots, where many samples show several-fold improvements.

**Negative gradients are not always harmful.** It is important to note that negative gradients are *not* inherently harmful. In terms of the likelihood change $\Delta(x)$, it is possible, as shown in Fig. 1c, that under `Pos Only` training, some samples exhibit even smaller $\Delta(x)$ than GRPO, which uses both positive and negative gradients. This decrease may result either from negative gradients "squeezing" the likelihood mass toward confident $y^+$ [20], or from certain negative gradients increasing $\Delta(x)$, as suggested by our Theorem 4.4 below. Moreover, in terms of performance, we demonstrate that `Pos Only` actually yields a 1.3% drop in average performance compared to GRPO. The result is shown in Tab. 2 for finetuning Qwen-Math-1.5B on the math dataset using GRPO and `Pos Only` respectively.

**What do negative gradients of questions with small likelihood change penalize?** We find that the minimal likelihood change occurs in correct responses of those questions in the dataset that have partially correct (but eventually incorrect) responses or have correct responses. In Fig. 2, we present two representative examples from Qwen-0.5B, where the incorrect responses are partially correct and the corresponding correct responses show the smallest likelihood change. The left example, highlighted in red, shows an answer that is nearly correct and corresponds to the red-boxed sample in Fig. 1a. The right example provides a correct answer but does not follow the expected output format (highlighted in red); it corresponds to the blue-boxed sample in Fig. 1a. Additional examples are provided in Appendix. This suggests that penalizing entire partially correct responses is suboptimal.

**Controlling negative gradients at the token level can improve performance.** While negative gradients do not necessarily hurt performance and penalizing entire incorrect responses is suboptimal, we show in Sec. 5 that carefully reducing negative gradients at *a token level* can mitigate small likelihood change and lead to improvement boosts. Our approach specifically leverages the previous observation that partial responses might be partly correct by filtering out and selectively penalizing their consistuent tokens.

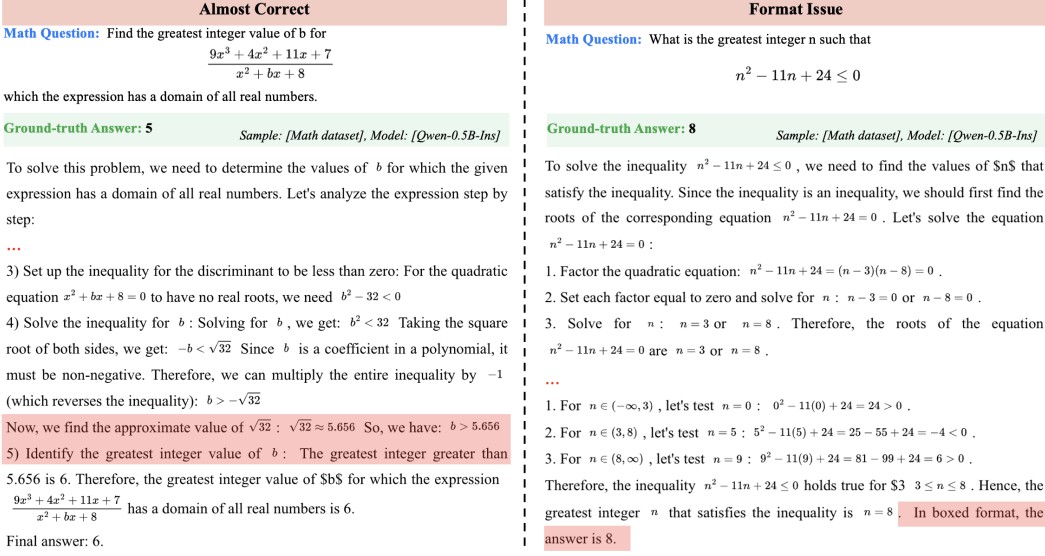

Figure 2: Inspecting negative (incorrect) samples of questions with small average likelihood change $\Delta(\boldsymbol{x})$ (Eq. (2)) reveals that they are either nearly correct (Left) or get the correct response in a wrong answer format (Right). Thus, penalizing entire negative sample responses might be suboptimal. Red dashed lines denote omitted reasoning steps.

## 4 Lazy Likelihood Displacement

We have shown that negative gradient in GRPO can cause a marginal or even reduced likelihood of correct responses $\boldsymbol{y}^+$. Here, we formalize this observation by introducing the concept of **Lazy Likelihood Displacement (LLD)**—a phenomenon in which the probability of correct answers decreases or increases only marginally after training. Our experiments reveal that LLD frequently emerges in group-based policy optimization settings, indicating that GRPO and its variants are also subject to the same limitations observed in preference optimization frameworks such as DPO [20, 19].

**Definition 4.1** *Let $\pi_{\theta_{init}}$ and $\pi_{\theta_{fin}}$ denote the initial and final language models, before and after optimizing a preference learning objective $\mathcal{J}$ (e.g., Eq. (1)) over a dataset $\mathcal{D}$, such that $\mathcal{J}(\theta_{fin}) < \mathcal{J}(\theta_{init})$. We say that **LLD** occurs for a tuple $(\mathbf{x}, \boldsymbol{y}^+) \in \mathcal{D}$ if, for small nonnegative constant $\epsilon \geq 0$,*

$$\ln \pi_{\theta_{fin}}(\boldsymbol{y}^+|\mathbf{x}) < \ln \pi_{\theta_{init}}(\boldsymbol{y}^+|\mathbf{x}) + \epsilon. \tag{3}$$

### 4.1 Understanding LLD

We begin by showing that GRPO effectively performs a weighted group preference optimization between two groups of responses: correct and incorrect ones (detailed proof in the appendix).

**Lemma 4.2** *When reward is binary, GRPO performs preference optimization between two distinct groups: the group of correct responses $(r_i = 1)$ and the group of incorrect responses $(r_i = 0)$. Specifically, the optimization objective reduces to the following:*

$$p^+ \sum_{i=1}^{N^+} \min\left(\frac{\pi_\theta(\boldsymbol{y}_i^+|\boldsymbol{x})}{\pi_{\theta_{old}}(\boldsymbol{y}_i^+|\boldsymbol{x})}, 1+\varepsilon\right) - p^- \sum_{j=1}^{N^-} \max\left(\frac{\pi_\theta(\boldsymbol{y}_j^-|\boldsymbol{x})}{\pi_{\theta_{old}}(\boldsymbol{y}_j^-|\boldsymbol{x})}, 1-\varepsilon\right), \tag{4}$$

*where $p \triangleq p(\boldsymbol{x}) \triangleq \frac{1}{G}\sum_{i\in[G]} \mathbb{1}[r_i(x) = 1]$ denotes the correctness rate for a given input, $N^+ = pG$ and $N^- = (1-p)G$ are the sizes of the correct and incorrect response groups respectively, and the group-specific weights are defined as $p^+ = \frac{1-p}{\sqrt{p(1-p)}}$ and $p^- = \frac{p}{\sqrt{p(1-p)}}$.*

Next, we introduce an assumption of unconstrained features, which allows us to focus our analysis on the final-layer hidden embedding $\mathbf{h_x} \in \mathbb{R}^d$ and the token unembedding matrix $\mathbf{W} \in \mathbb{R}^{|\mathcal{V}|\times d}$, where $\mathcal{V}$ is the vocabulary of tokens.

**Assumption 4.3 (Unconstrained Features)** *Expressive (enough) neural networks can produce unconstrained embeddings $\mathbf{h_x} \in \mathbb{R}^d$ independent of the architecture's specific complexities [27, 16, 34, 19]. These embeddings are subsequently transformed into logits by a token unembedding matrix $\mathbf{W} \in \mathbb{R}^{|\mathcal{V}| \times d}$. The resulting logits are passed through a softmax function to yield a probability distribution over possible next tokens. To assign probabilities to sequences $\mathbf{y} \in \mathcal{V}^*$, the language model $\pi_\theta$ operates in an autoregressive manner, i.e., $\pi_\theta(\mathbf{y} \mid \mathbf{x}) = \prod_{k=1}^{|\mathbf{y}|} \mathrm{Softmax}(\mathbf{W}\mathbf{h_{x,y_{<k}}})_{y_k}$.*

Here, $\mathbf{h_{x,y_{<k}}}$ is the embedding of sequence $(\boldsymbol{x}, \boldsymbol{y}_{<k})$ and $\mathrm{Softmax}(\cdot)_{y_k}$ is the $y_k$-th entry of the $V$-dimensional softmax map. We focus on the online training setting for GRPO, as state in Sec. 2.2, the effect of the clipping operation is effectively removed. We assume that the objective is optimized via gradient flow and theoretically analyze the dynamics of the log-likelihood of a positive response, $\frac{d}{dt} \ln \pi_{\theta(t)}(\mathbf{y}_i^+ \mid \mathbf{x})$, yielding the following characterization[1] (see the appendix for a proof).

**Theorem 4.4** *For any question $\boldsymbol{x}$, at any time $t \geq 0$ of training, and any correct response $\boldsymbol{y}_i^+, i \in [N^+]$, in addition to the dependence on token unembeddings, the likelihood change $\frac{d}{dt} \ln \pi_{\theta(t)}(\boldsymbol{y}_i^+|\mathbf{x})$ exhibits increased laziness (that is, has smaller magnitude) as the following quantity increases:*

$$\underbrace{p^- \sum_{k=1}^{|\boldsymbol{y}_i^+|} \sum_{j=1}^{N^-} \sum_{k'=1}^{|\boldsymbol{y}_j^-|} \alpha_{k,k'}^- \cdot \langle \mathbf{h_{x,y_{i,<k}^+}}, \mathbf{h_{x,y_{j,<k'}^-}} \rangle}_{\textit{impact of negative gradient}} - p^+ \sum_{k=1}^{|\boldsymbol{y}_i^+|} \sum_{i'=1}^{N^+} \sum_{k''=1}^{|\boldsymbol{y}_{i'}^+|} \alpha_{k,k''}^+ \cdot \langle \mathbf{h_{x,y_{i,<k}^+}}, \mathbf{h_{x,y_{i',<k''}^+}} \rangle. \quad (5)$$

*Here, $\alpha_{k,k'}^-$ and $\alpha_{k,k''}^+$ are token-level prediction error similarity weights, which quantify the similarity of token-level prediction error across responses (see Appendix for formal definitions. ). We refer to the quantity in Eq. (5) as Group Weighted Hidden Embedding Score (GWHES).*

The first term in Eq. (5) captures the influence of negative gradients on the likelihood of the correct response. Specifically, this shows that a negative token with a large value of $\alpha_{k,k'}^- \cdot \langle \mathbf{h_{x,y_{i,<k}^+}}, \mathbf{h_{x,y_{j,<k'}^-}} \rangle$ is more likely to cause the LLD of the correct responses. This motivates the following corollary.

**Corollary 4.5** *For any question $\mathbf{x}$, the negative gradient associated with the hidden embedding of a token $k'$ in any incorrect response $\mathbf{y}_j^-$, $j \in [N^-]$, will exert a greater adverse effect on the likelihood change $\frac{d}{dt} \ln \pi\theta(t)(\mathbf{y}_i^+|\mathbf{x})$ as the following quantity getslarger:*

$$\sum_{k=1}^{|\boldsymbol{y}_i^+|} \alpha_{k,k'}^- \cdot \langle \mathbf{h_{x,y_{i,<k}^+}}, \mathbf{h_{x,y_{j,<k'}^-}} \rangle. \quad (6)$$

This can occur when the negative and positive tokens have high embedding similarity, theoretically supporting the observation in Fig. 2 that LLD samples are often nearly correct.

## 4.2 Identifying LLD Samples

Theorem 4.4 suggests using $\Delta_{\mathrm{GWHES}}$ as a metric to identify samples that cause LLD. Here, we verify the validity of this metric experimentally. For Qwen-1.5B-deepseek, we generate responses for the first 100 questions from the AIME dataset (1983–2023) and compute the GWHES scores for those questions with non-extreme predicted probabilities ($0 < p < 1$), resulting in 57 valid questions. For Qwen-2.5Math-1.5B, we use questions from the Math dataset and retain 47 valid questions. We then

| Top-K | Qwen-1.5B-deepseek | | Qwen-1.5B-math | |
|---|---|---|---|---|
| | GWHES | Random | GWHES | Random |
| 10 | 50% | 17.5% | 60% | 21.3% |
| 15 | 75% | 26.3% | 75% | 31.9% |

Table 1: Ranking questions by GWHES results in a significantly higher Top-$K$ overlap with the likelihood-change ranking compare to ranking randomly. This demonstrates the effectiveness of GWHES in identifying LLD samples.

calculate the Top-$K$ overlap accuracy between two rankings of questions: one based on the ascending

---

[1]Theorem 4.4 can be seen as an extension of [19, Thm.3] for DPO to the more challenging GRPO setting with group-based structure. Unlike DPO, we will exploit this group-structure to selectively penalize tokens rather than responses.

order of likelihood changes $\Delta(\boldsymbol{x})$ as defined in Eq. (2) (denoted $\mathcal{Q}_K^L$), and the other based on the descending order of GWHES scores as defined in Eq. (5) (denoted $\mathcal{Q}_K^S$). The overlap accuracy is computed as $\frac{|Q_K^L \cap Q_K^S|}{K}$. The results in Tab. 1 indicate that the overlap achieved using GWHES is significantly greater than that of the random ranking baseline, highlighting a strong alignment between high GWHES scores and LLD samples.

## 5 Selective Token Penalization in Incorrect Responses

Theorem 4.4 identifies samples prone to causing LLD and Corollary 4.5 quantifies the influence of individual tokens' negative gradients on the likelihood of correct responses. The goal of this section is to explore whether appropriately controlling negative gradients can mitigate LLD and further lead to performance improvements over vanilla GRPO. One approach could be using $\Delta_{\text{GWHES}}$ to identify negative samples $\boldsymbol{y}_i^-$ that cause LLD and entirely discard them during training. However, this compromises data efficiency and fails to address cases where samples only marginally improve $\boldsymbol{y}^+$. Instead, we propose a more targeted approach: identify tokens that negatively impact $\Delta(\boldsymbol{x})$ and reduce the penalty applied to them.

**Math Question:** How many elements are in the intersection of the set of all the prime numbers less than 30 and the set of all the odd numbers greater than zero?

**Ground-truth Answer: 9**                                                  *Sample from [Math dataset]*

**Correct Response**

To determine **...** Step 1: Identify the set of all prime numbers less than 30. The prime numbers less than 30 are: 2, 3, 5, 7, 11, 13, 17, 19, 23, 29. So, the set of all prime numbers less than 30 is {2, 3, 5, 7, 11, 13, 17, 19, 23, 29}.
Step 2: Identify the set of all odd numbers greater than zero. The set of all odd numbers greater than zero is: {1, 3, 5, 7, 9, 11, 13, 15, 17, 19, 21, 23, 25, 27, 29, ...}. So, the set of all odd numbers greater than zero is {1, 3, 5, 7, 9, 11, 13, 15, 17, 19, 21, 23, 25, 27, 29, ...}.
Step 3: Find the intersection of the two sets **...** From the two sets, we see that the common elements are 3, 5, 7, 11, 13, 17, 19, 23, and 29. So, the intersection is {3, 5, 7, 11, 13, 17, 19, 23, 29}.
Step 4: Count the number of elements in the intersection. The number of elements in the intersection is 9. Therefore, the number of elements in the intersection of the set of all the prime numbers less than 30 and the set of all the odd numbers greater than zero is \boxed{9}.

**Incorrect Response** Contain Correct Steps

To determine **...** First, let 's list the prime numbers less than 3 0 : 2 , 3 , 5 , 7 , 1 1 , 1 3 , 1 7 , 1 9 , 2 3 , 2 9 So, the set of all prime numbers less than 3 0 is : { 2 , 3 , 5 , 7 , 1 1 , 1 3 , 1 7 , 1 9 , 2 3 , 2 9 }
Next, let 's list the set of all odd numbers greater than zero . The set of all odd numbers greater than zero is : { 1 , 3 , 5 , 7 , 9 , 1 1 , 1 3 , 1 5 , 1 7 , 1 9 , 2 1 , 2 3 , 2 5 , 2 7 } Now, we find the intersection of these two sets . The intersection consists of numbers that are both in the set of prime numbers less than 3 0 and in the set of all odd numbers greater than zero . From the lists above , we can see that the only numbers that are both prime and odd are the odd prime numbers .
Therefore, the intersection is : { 3 , 5 , 7 , 1 1 , 1 3 , 1 7 , 1 9 , 2 3 } To find the number of elements in the intersection , we count the elements in the list : { 3 , 5 , 7 , 1 1 , 1 3 , 1 7 , 1 9 , 2 3 } There are 8 elements in the intersection . Thus , the number of elements in the intersection is \boxed{ 8 }.

Figure 3: Key insight: Tokens of negative samples (incorrect responses) can be logically or step-correct. Tokens with high NTHR tend to strongly correlate with these types of tokens (highlighted in red). yThe bold dots represent omitted reasoning.

### 5.1 Negative Token Hidden Reward (NTHR)

Motivated by Corollary 4.5, we define the impact that an individual token $k'$ of incorrect response $\boldsymbol{y}_j^-, j \in [N^-]$ has on the likelihood of a group of correct responses as follows:

$$s_{j,<k'}^- := \sum_{i=1}^{N^+} \sum_{k=1}^{|\mathbf{y}_i^+|} \alpha_{k,k'}^- \cdot \left\langle \mathbf{h}_{\mathbf{x}, \mathbf{y}_{i,<k}^+}, \mathbf{h}_{\mathbf{x}, \mathbf{y}_{j,<k'}^-} \right\rangle, \qquad (7)$$

which defines a relative influence from imposing a negative gradient on $\boldsymbol{y}_j^-$ to all tokens in all positive responses. Specifically, a token from an incorrect response can negatively affect the likelihood of a group of a group of correct responses when $s_{j,<k'}^- \geq 0$, with the magnitude of $s_{j,<k'}^-$ indicating the severity of this effect. We refer to this value as the **negative token hidden reward** (NTHR):

it quantifies a token's detrimental impact on correct responses' likelihood. In Fig. 3, we visualize tokens with high $s^-_{j,<k'}$ values—highlighted in red. We find that many of these influential tokens are logically or stepwise correct terms (such as "odd," "prime," and "intersection" in the specific example). These terms exhibit strong semantic alignment with the correct responses.

## 5.2  NTHR selective token penalization.

As shown in Corollary 4.5, tokens with high positive NTHR values tend to cause significant reductions in the likelihood of generated correct responses. To mitigate this adverse effect, we introduce a selective penalization strategy (detailed in Algorithm 1 and complexity discussion in Appendix) that attenuates the penalty on tokens from negative responses whose NTHR scores exceed a threshold $\tau$. Concretely, we define the resulting set of selected tokens as:

$$\mathbf{V}^-_j = \left\{ \mathbf{y}^-_{j,k'} \,\middle|\, s^-_{j,<k'} > \tau \right\}. \tag{8}$$

To determine the threshold $\tau$, an effective pratical strategy is computing the minimum average token-level influence exerted by each correct response on all other correct responses. Formally, we set $\tau = \beta \cdot \min_{i' \in [N^+]} \bar{s}^+_{i'}$,

---

**Algorithm 1** NTHR selective token penalization

1: **Input:** Responses' hidden embeddings $\{\mathbf{h}^+_i\}^{N^+}_{i=1}$, $\{\mathbf{h}^-_j\}^{N^-}_{j=1}$, coefficient $\alpha$, scale factor $\eta$ and $\beta$.
2: **for** $i' = 1$ to $N^+$ **do**
3: $\quad \bar{s}^+_{i'} \leftarrow$ Eq. (7)
4: **end for**
5: $\tau \leftarrow \beta \cdot \min_{i' \in [N^+]} \bar{s}^+_{i'}$
6: **Select negative tokens:**
7: **for** each negative response $j$ **do**
8: $\quad \mathbf{V}^-_j \leftarrow \left\{ \mathbf{y}^-_{j,k'} \mid s^-_{j,<k'} > \tau \right\}$
9: **end for**
10: **Apply selective penalization:**
11: **for** each retained token $\mathbf{y}^-_{j,k'} \in \mathbf{V}^-_j$ **do**
12: $\quad \hat{A}^-_{j,k',\eta} \leftarrow \eta \cdot \hat{A}^-_{j,k'}$
13: **end for**

---

where $\beta$ is a scale factor and $\bar{s}^+_{i'}$ measures the average impact of the $i'$-th correct response's tokens on the likelihoods of other correct responses:

$$\bar{s}^+_{i'} := \frac{1}{|\mathbf{y}^+_{i'}|} \sum^{|\mathbf{y}^+_{i'}|}_{k''=1} \sum^{N^+}_{i=1} \sum^{|\mathbf{y}^+_i|}_{k=1} \alpha^+_{k,k''}(t) \cdot \left\langle \mathbf{h}_{\mathbf{x},\mathbf{y}^+_{i,<k}}(t), \mathbf{h}_{\mathbf{x},\mathbf{y}^+_{i',<k''}}(t) \right\rangle. \tag{9}$$

A larger $\bar{s}^+_{i'}$ indicates a stronger mutual influence among positive responses, which could be interpreted as an estimation of the "local elasticity" of a deep neural network [6]. Finally, to apply selective penalization, we define the advantage of each retained token using a scale factor $\eta < 1$, resulting in $\hat{A}^-_{j,k',\eta} := \eta \cdot \hat{A}^-_{j,k'}$. The scale factor will reduce the penalty on selected negative tokens.

## 5.3  NTHR Selective Token Penalization Mitigates LLD

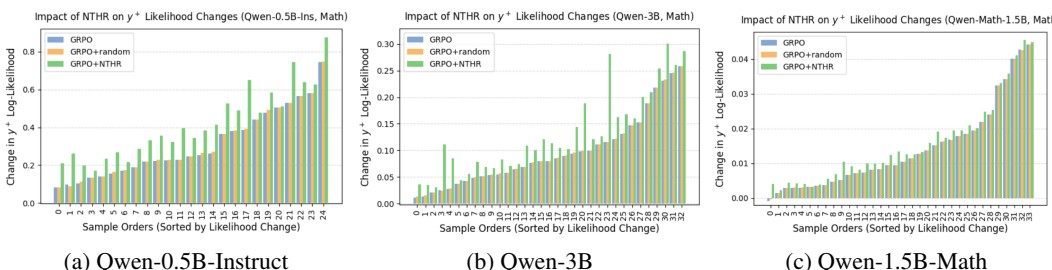

(a) Qwen-0.5B-Instruct $\qquad$ (b) Qwen-3B $\qquad$ (c) Qwen-1.5B-Math

Figure 4: GRPO+NTHR consistently improves likelihood change of correct responses, as indicated by the green bars exceeding the blue bars. While GRPO+Random offers only modest improvements, GRPO+NTHR consistently outperforms it, highlighting the effectiveness of NTHR in identifying LLD tokens.

To evaluate the effectiveness of our NTHR selective token penalization in mitigating LLD, we conduct experiments using the same setup described in Sec. 4. Specifically, we identify the selected

token set $\mathbf{V}_j^-$ in incorrect responses using a scale parameter $\beta = 1$, and apply no penalization to these tokens by setting $\eta = 0$. We refer to this variant as GRPO+NTHR (green bars). We then compare the following two baselines: (1) Standard GRPO (blue bars), and (2) GRPO+Random (orange bars), where we randomly select $|\mathbf{V}_j^-|$ tokens and set their advantage to 0 (thus avoid penalizing them in optimization).

The results in Fig. 4 show that GRPO+NTHR consistently improves likelihood change across all samples, as evidenced by the green bars surpassing the blue bars. Notably, while removing negative gradients entirely—as in the `Pos Only` in Fig. 1c—can sometimes reduce the likelihood change compared to GRPO in Qwen-Math-1.5B, our GRPO+NTHR approach (Fig. 4c) maintains consistent improvements. Furthermore, unlike GRPO+Random that provides only modest gains across all models, GRPO+NTHR delivers consistent and substantial improvements, highlighting the effectiveness of our proposed NTHR in identifying LLD tokens.

## 5.4 Performance of NTHR Selective Token Penalization

Having demonstrated the impact of negative gradients on likelihood change and the effectiveness of our method in mitigating the LLD issue in Sec. 5.3, we now study the effect of negative gradient on model performance. For this, we finetune models with different sizes using MATH dataset (levels 3–5) [7] and use greedy decoding to evaluate finetuned models on five math benchmarks: AIME24, AMC, MATH500, Minerva, and Olympiad (detailed implementation details in Appendix).

We find that NTHR consistently boosts model performance across various sizes, with detailed results presented in Tab. 2. Notably, even for Qwen2.5-Math which shows a substantial improvement of over 20% after RL fine-tuning—outpacing gains seen in other models—NTHR still provides an average improvement of 0.8%. For Qwen2.5-0.5B-Ins and Qwen2.5-1.5B-Ins, NTHR consistently outperforms GRPO by 1.1% and 1.5% on average respectively. Similar improvements are observed with a larger model Qwen2.5-3B, where NTHR reaches 36.30 % compared to GRPO's 33.88 % and the base model's 31.36 %. We further fine-tune Qwen2.5-Math on the more challenging DeepScaler [14] dataset and observe that NTHR outperforms GRPO by 1.8%.

| Base model + Method | AIME24 | AMC | MATH500 | Minerva | Olympiad | Avg. |
|---|---|---|---|---|---|---|
| **Qwen2.5-Math-1.5B** | | | | | | |
| Base | 3.3 | 20.0 | 39.6 | 7.7 | 24.9 | 19.10 |
| GRPO | 13.3 | 57.5 | **71.8** | 29.0 | 34.1 | 41.14 |
| Pos Only | 10.0 | 57.5 | 70.6 | 30.1 | 31.0 | 39.84 |
| NTHR | **16.7** | 57.5 | 70.8 | **30.5** | **34.2** | **41.94** |
| **Qwen2.5-0.5B-Ins** | | | | | | |
| Base | 0.0 | 2.5 | 33.4 | 4.4 | 7.0 | 9.46 |
| GRPO | 0.0 | 7.5 | 33.8 | **9.2** | 8.1 | 11.72 |
| NTHR | 0.0 | **10.0** | **36.6** | 8.1 | **8.6** | **12.66** |
| **Qwen2.5-1.5B-Ins** | | | | | | |
| Base | 0.0 | 22.5 | 53.0 | 19.1 | 20.7 | 23.06 |
| GRPO | 3.3 | 32.5 | 57.2 | 18.8 | **23.0** | 26.96 |
| NTHR | **6.7** | **35.0** | **58.8** | **21.0** | 20.9 | **28.48** |
| **Qwen2.5-Math-1.5B (deepscaler)** | | | | | | |
| Base | 3.3 | 20.0 | 39.6 | 7.7 | 24.9 | 19.10 |
| GRPO | 10.0 | 42.5 | 72.4 | **32.4** | **31.9** | 37.80 |
| NTHR | **16.7** | **47.5** | **73.2** | 29.4 | 31.4 | **39.60** |
| **Qwen2.5-3B** | | | | | | |
| Base | 10.0 | 37.5 | 58.6 | 26.1 | 24.6 | 31.36 |
| GRPO | 6.7 | 35.0 | **66.6** | 31.2 | **29.9** | 33.88 |
| NTHR | **10.0** | **47.5** | 65.6 | **31.6** | 26.8 | **36.30** |

Table 2: Results across selected math benchmarks for different Qwen2.5 models and methods. NTHR consistently provides average performance gains on various models.

## 6 Conclusion

In this work, we study how negative gradients affect the likelihood of correct responses in GRPO. We identify a phenomenon we term Lazy Likelihood Displacement (LLD), where penalization of

incorrect responses inadvertently reduces or lead to small likelihood change of correct ones. To address this, we propose the negative token hidden reward (NTHR) along with a selective token-level penalization strategy. NTHR reduces the penalty on tokens in incorrect responses that contribute most to lowering the likelihood of correct responses, thus successfully mitigating the LLD issue. Through experiments on math reasoning tasks across models ranging from 0.5B to 3B, we demonstrate that NTHR improves GRPO, validating the effectiveness of our approach in addressing LLD and enhancing performance.

**Acknowledgments:** This work was partially funded by the NSERC Discovery Grant RGPIN-2021-03677, Alliance Grant ALLRP 581098-22, the Natural Science and Engineering Research Council of Canada (NSERC), the Canada CIFAR AI Chairs program, the Canada Research Chair program, an IITP grant funded by MSIT, the BC DRI Group, and the Digital Research Alliance of Canada.

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

# 7 Appendix

**Notation.** For any time $t \geq 0$, we use $\boldsymbol{W}(t)$, $\boldsymbol{w}_z(t)$, and $\boldsymbol{h}_z(t)$ to denote the token unembedding matrix, unembedding of a token $z \in \mathcal{V}$, and hidden embedding of $z \in \mathcal{V}^*$, respectively. We let $\boldsymbol{z}_k$ be the $k$-th token in $\boldsymbol{z}$ and $\boldsymbol{z}_{<k}$ be the first $k-1$ tokens in $\boldsymbol{z}$. For a question $\boldsymbol{x}$, the old policy $\pi_{\theta_{\text{old}}}$ generates a group of $G$ samples and resulting $(\boldsymbol{x}, \{\boldsymbol{y}_i^+\}_{N^+}, \{\boldsymbol{y}_j^-\}_{N^-})$, where $N^+ + N^- = G$. Lastly, we denote by $\boldsymbol{e}_z \in \mathbb{R}^{|\mathcal{V}|}$ the standard basis vector corresponding to $z \in \mathcal{V}$.

## 7.1 Proof of Lemma 4.2: GRPO as Group Preference Optimization

In this section, we demonstrate that training with GRPO constitutes preference optimization. We adopt a binary reward system, assigning $r = 1$ to correct responses and $r = 0$ to incorrect ones, consistent with recent works [13, 29]. It is notable that this also applies to its variants, e.g., DAPO [29], Dr.GRPO [13], GPG [1], etc.

For a single question $\boldsymbol{x}$, we simplify the expected loss $\mathbb{E}_{\{\boldsymbol{y}_i\}_{i=1}^G \sim \pi_{\theta_{\text{old}}}(\cdot|\boldsymbol{x})}[\mathcal{J}_{\boldsymbol{x}}]$ in Eq. (1) by omitting the token normalization term $\frac{1}{\sum_{i=1}^G |\boldsymbol{y}_i|}$, yielding:

$$\mathbb{E}_{\{\boldsymbol{y}_i\}_{i=1}^G \sim \pi_{\theta_{\text{old}}}(\cdot|\boldsymbol{x})}\left[\sum_{i=1}^G \sum_{k=1}^{|\boldsymbol{y}_i|} \min\left(\gamma_{i,k}(\theta)\hat{A}_{i,k}, \hat{A}_{i,k} \cdot \text{clip}\left(\gamma_{i,k}(\theta), 1-\varepsilon, 1+\varepsilon\right)\right)\right]. \tag{10}$$

The success probability for question $\boldsymbol{x}$ is $p = \hat{\mathbb{P}}_i(r_{i,\boldsymbol{x}} = 1) \approx \mathbb{P}(r_{\boldsymbol{x}} = 1)$ where we use a population approximation for large enough $G$. In this case, we take $\mu = p$ and $\sigma = \sqrt{p(1-p)}$. Consequently, the advantage $\hat{A}_{i,k}$ becomes:

$$\hat{A}_{i,k} = \begin{cases} \frac{1-p}{\sqrt{p(1-p)}} & \text{if } r_i = 1, \\ -\frac{p}{\sqrt{p(1-p)}} & \text{if } r_i = 0. \end{cases} \tag{11}$$

Since

$$\min\left(\gamma_{i,k}(\theta)\hat{A}_{i,k}, \; \hat{A}_{i,k} \cdot \text{clip}\left(\gamma_{i,k}(\theta), \; 1-\varepsilon, \; 1+\varepsilon\right)\right)$$

is equivalent tos

$$\begin{cases} \hat{A}_{i,k} \cdot \min\left(\gamma_{i,k}(\theta), \; 1+\varepsilon\right), & \text{if } \hat{A}_{i,k} > 0 \Leftrightarrow r_i = 1, \\ \hat{A}_{i,k} \cdot \max\left(\gamma_{i,k}(\theta), \; 1-\varepsilon\right), & \text{if } \hat{A}_{i,k} < 0 \Leftrightarrow r_i = 0, \end{cases}$$

the expected loss then becomes:

$$\begin{aligned} \mathbb{E}_{\{\boldsymbol{y}_i\}_{i=1}^G \sim \pi_{\theta_{\text{old}}}(\cdot|\boldsymbol{x})}[\mathcal{J}_{\boldsymbol{x}}] = {}& \frac{1-p}{\sqrt{p(1-p)}} \mathbb{E}_{\{\boldsymbol{y}_i\}_{i=1}^G \sim \pi_{\theta_{\text{old}}}(\cdot|\boldsymbol{x})}\left[\min\left(\frac{\pi_\theta(\boldsymbol{y}_i|\boldsymbol{x})}{\pi_{\theta_{\text{old}}}(\boldsymbol{y}_i|\boldsymbol{x})}, 1+\varepsilon\right)\mathbf{1}_{r_i=1}\right] \\ & - \frac{p}{\sqrt{p(1-p)}} \mathbb{E}_{\{\boldsymbol{y}_i\}_{i=1}^G \sim \pi_{\theta_{\text{old}}}(\cdot|\boldsymbol{x})}\left[\max\left(\frac{\pi_\theta(\boldsymbol{y}_i|\boldsymbol{x})}{\pi_{\theta_{\text{old}}}(\boldsymbol{y}_i|\boldsymbol{x})}, 1-\varepsilon\right)\mathbf{1}_{r_i=0}\right]. \end{aligned} \tag{12}$$

When generating $G$ sampled responses, the loss becomes:

$$p^+ \sum_{i=1}^{N^+} \min\left(\frac{\pi_\theta(\boldsymbol{y}_i^+|\boldsymbol{x})}{\pi_{\theta_{\text{old}}}(\boldsymbol{y}_i^+|\boldsymbol{x})}, 1+\varepsilon\right) - p^- \sum_{j=1}^{N^-} \max\left(\frac{\pi_\theta(\boldsymbol{y}_j^-|\boldsymbol{x})}{\pi_{\theta_{\text{old}}}(\boldsymbol{y}_j^-|\boldsymbol{x})}, 1-\varepsilon\right), \tag{13}$$

where $N^+ = pG$, $N^- = (1-p)G$, $p^+ = \frac{1-p}{\sqrt{p(1-p)}}$ and $p^- = \frac{p}{\sqrt{p(1-p)}}$. This represents group preference optimization, i.e., increasing the likelihood of correct responses while penalizing incorrect ones.

## 7.2 Proof of Theorem 4.4

Assume that all responses are sequences whose first tokens are distinct from each other. Then we analyze the likelihood change for a correct response $\boldsymbol{y}_i^+$ by measuring $\frac{d}{dt} \ln \pi_{\theta(t)}(\mathbf{y}_i^+|\mathbf{x})$. According

to the chain rule, using the loss in Eq. (13) and denoting $\pi_{<k} = \pi_\theta(\mathbf{y}_{i,k}^+|\mathbf{x}, \mathbf{y}_{i,<k}^+)$ and $\pi_{\text{old},<k} = \pi_{\theta_{\text{old}}}(\mathbf{y}_{i,k}^+|\mathbf{x}, \mathbf{y}_{i,<k}^+)$ for brevity , we can obtain:

$$\frac{d}{dt}\ln\pi_{\theta(t)}(\boldsymbol{y}_i^+|\boldsymbol{x}) = \left\langle \nabla\ln\pi_{\theta(t)}(\boldsymbol{y}_i^+|\boldsymbol{x}), \frac{d\theta(t)}{dt} \right\rangle \tag{14}$$

$$= \left\langle \nabla\ln\pi_{\theta(t)}(\boldsymbol{y}_i^+|\boldsymbol{x}), \right.$$

$$p^+ \sum_{i'=1}^{N^+}\sum_{k=1}^{|\boldsymbol{y}_{i'}^+|} \frac{\pi_{<k}}{\pi_{\text{old},<k}}\delta\left(1+\epsilon-\frac{\pi_{<k}}{\pi_{\text{old},<k}}\right)\nabla\ln\pi_{\theta(t)}(\boldsymbol{y}_{i',k}^+|\boldsymbol{x},\boldsymbol{y}_{i',<k}^+)$$

$$\left. -p^- \sum_{j=1}^{N^-}\sum_{k'=1}^{|\boldsymbol{y}_j^-|} \frac{\pi_{<k'}}{\pi_{\text{old},<k'}}\delta\left(\frac{\pi_{<k'}}{\pi_{\text{old},<k'}}-1+\epsilon\right)\nabla\ln\pi_{\theta(t)}(\boldsymbol{y}_{j,k'}^-|\boldsymbol{x},\boldsymbol{y}_{j,<k'}^-) \right\rangle \tag{15}$$

where $\delta(x) = \begin{cases} 1 & \text{if } x \geq 0 \\ 0 & \text{if } x < 0 \end{cases}$ is the gradient of the clip function. We assume $\pi_{\theta(t)} = \pi_{\text{old}}$ since GRPO is online or nearly online due to sampling new responses at each iteration. Thus we obtain:

$$\frac{d}{dt}\ln\pi_{\theta(t)}(\boldsymbol{y}_i^+|\boldsymbol{x}) = \left\langle \nabla\ln\pi_{\theta(t)}(\boldsymbol{y}_i^+|\boldsymbol{x}), p^+\sum_{i'=1}^{N^+}\nabla\ln\pi_{\theta(t)}(\boldsymbol{y}_{i'}^+|\boldsymbol{x}) - p^-\sum_{j=1}^{N^-}\nabla\ln\pi_{\theta(t)}(\boldsymbol{y}_j^-|\boldsymbol{x}) \right\rangle.$$

As per the unconstrained features Assumption 4.3, the model's trainable parameters are

$$\theta = \left(\boldsymbol{W}, \mathbf{h}_{\boldsymbol{x}}, \left\{\mathbf{h}_{\boldsymbol{x},\boldsymbol{y}_{i',<k}^+}\right\}_{i'\in[N^+],k\in\{2,\ldots,|\boldsymbol{y}_{i'}^+|\}}, \left\{\mathbf{h}_{\boldsymbol{x},\boldsymbol{y}_{j,<k'}^-}\right\}_{j\in[N^-],k'\in\{2,\ldots,|\boldsymbol{y}_j^-|\}}\right).$$

Here, we also used the (mild) assumption that all responses differ in their first token. Unfolding the gradients with respect to these parameters yields:

$$\frac{d}{dt}\ln\pi_{\theta(t)}(\boldsymbol{y}_i^+|\boldsymbol{x}) = \left\langle \nabla_{\boldsymbol{W}}\ln\pi_{\theta(t)}(\boldsymbol{y}_i^+|\boldsymbol{x}), p^+\sum_{i'=1}^{N^+}\nabla_{\boldsymbol{W}}\ln\pi_{\theta(t)}(\boldsymbol{y}_{i'}^+|\boldsymbol{x}) - p^-\sum_j^{N^-}\nabla_{\boldsymbol{W}}\ln\pi_{\theta(t)}(\boldsymbol{y}_j^-|\boldsymbol{x}) \right\rangle$$

$$+ \left\langle \nabla_{\mathbf{h}_{\mathbf{x}}}\ln\pi_{\theta(t)}(\boldsymbol{y}_{i,1}^+|\boldsymbol{x}), p^+\sum_{i'=1}^{N^+}\nabla_{\mathbf{h}_{\mathbf{x}}}\ln\pi_{\theta(t)}(\boldsymbol{y}_{i',1}^+|\boldsymbol{x}) - p^-\sum_{j=1}^{N^-}\nabla_{\mathbf{h}_{\mathbf{x}}}\ln\pi_{\theta(t)}(\boldsymbol{y}_{j,1}^-|\boldsymbol{x}) \right\rangle$$

$$+ \sum_{k=2}^{|\boldsymbol{y}_i^+|}||\nabla_{\mathbf{h}_{\boldsymbol{x},\boldsymbol{y}_{i,<k}^+}}\ln\pi_{\theta(t)}(\boldsymbol{y}_{i,k}^+|\boldsymbol{x},\boldsymbol{y}_{i,<k}^+)||^2. \tag{16}$$

For softmax model output the gradients can be easily computed as follows:

$$\nabla_{\boldsymbol{W}}\ln\pi_{\theta(t)}(\boldsymbol{z}|\boldsymbol{x}) = \sum_{k=1}^{|\boldsymbol{z}|}\left(\mathbf{e}_{\boldsymbol{z}_k} - \pi_{\theta(t)}(\cdot|\boldsymbol{x},\boldsymbol{z}_{<k})\right)\mathbf{h}_{\boldsymbol{z}_{<k}}^\top(t)$$

In addition, the gradient with respect to the hidden representation at each position $k$ is:

$$\nabla_{\mathbf{h}_{\boldsymbol{x},\boldsymbol{z}_{<k}}}\ln\pi_{\theta(t)}(\boldsymbol{z}|\boldsymbol{x}) = \boldsymbol{w}_{\boldsymbol{z}_k}(t) - \sum_{z\in\mathcal{V}}\pi_{\theta(t)}(z|\mathbf{x},\boldsymbol{z}_{<k})\cdot\boldsymbol{w}_z(t), \quad k\in\{1,\ldots,|\boldsymbol{z}|\}$$

Putting this back in (16) together with a few algebra steps, yields

$$\frac{d}{dt}\ln\pi_{\theta(t)}(\boldsymbol{y}_i^+|\boldsymbol{x}) = \text{(I)} - \text{(II)} + \text{(III)} + \text{(IV)} \tag{17}$$

where:

$$(\text{I}) = p^+ \sum_{k=1}^{|\boldsymbol{y}_i^+|} \sum_{i'=1}^{N^+} \sum_{k''=1}^{|\boldsymbol{y}_{i'}^+|} \alpha_{k,k''}^+(t) \cdot \left\langle \mathbf{h}_{\boldsymbol{x},\boldsymbol{y}_{i,<k}^+}(t), \mathbf{h}_{\boldsymbol{x},\boldsymbol{y}_{i',<k''}^+}(t) \right\rangle \tag{18}$$

$$(\text{II}) = p^- \sum_{k=1}^{|\boldsymbol{y}_i^+|} \sum_{j=1}^{N^-} \sum_{k'=1}^{|\boldsymbol{y}_j^-|} \alpha_{k,k'}^-(t) \cdot \left\langle \mathbf{h}_{\boldsymbol{x},\boldsymbol{y}_{i,<k}^+}(t), \mathbf{h}_{\boldsymbol{x},\boldsymbol{y}_{j,<k'}^-}(t) \right\rangle \tag{19}$$

$$(\text{III}) = \left\langle \boldsymbol{w}_{\boldsymbol{y}_{i,1}^+}(t) - \sum_{z \in \mathcal{V}} \pi_{\theta(t)}(z|\boldsymbol{x}) \cdot \boldsymbol{w}_z(t), \sum_{i'=1}^{N^+} p^+ \boldsymbol{w}_{\boldsymbol{y}_{i',1}^+} - \sum_{j=1}^{N^-} p^- \boldsymbol{w}_{\boldsymbol{y}_{j,1}^-} \right\rangle \tag{20}$$

$$(\text{IV}) = \sum_{k=2}^{|\boldsymbol{y}^+|} \left\| \boldsymbol{w}_{y_{i,k}^+}(t) - \sum_{z \in \mathcal{V}} \pi_{\theta(t)}(z|\boldsymbol{x}, \boldsymbol{y}_{i,<k}^+) \cdot \boldsymbol{w}_z(t) \right\|^2 \tag{21}$$

where $\alpha_{k,k''}^+(t) = \left\langle \mathbf{e}_{\boldsymbol{y}_{i,k}^+} - \pi_{\theta(t)}(\cdot|\boldsymbol{x}, \boldsymbol{y}_{i,<k}^+), \mathbf{e}_{\boldsymbol{y}_{i',k''}^+} - \pi_{\theta(t)}(\cdot|\boldsymbol{x}, \boldsymbol{y}_{i',<k''}^+) \right\rangle$ and $\alpha_{k,k'}^-(t) = \left\langle \mathbf{e}_{y_{i,k}^+} - \pi_{\theta(t)}(\cdot|\mathbf{x}, \mathbf{y}_{i,<k}^+), \mathbf{e}_{\boldsymbol{y}_{j,k'}^-} - \pi_{\theta(t)}(\cdot|\boldsymbol{x}, \boldsymbol{y}_{j,<k'}^-) \right\rangle$.

Specifically, (I) and (II) capture how token embeddings influence changes in likelihood, while (III) and (IV) reflect how the geometry of token unembeddings governs such changes. Our focus is on (I) and (II) because:

- Token embeddings encapsulate the contribution of all network parameters excluding the token unembedding layer.
- Token embeddings are influenced by the words in the sample, which span a broader space than token unembeddings.

Our results in Tab. 1 further validate the effectiveness of these terms. Thus we arrive at Theorem 4.4: as (II) − (I) increases, the likelihood change decreases.

## 7.3 Implementation Details

**Dataset.** For training, we use the MATH dataset (levels 3–5) to train the model. Additionally, we include a subset of the DeepScaler dataset [14], which contains more challenging problems. For evaluation, we assess the reasoning capabilities of the fine-tuned models on five standard math benchmarks: AIME 2024 [24], AMC, MATH500, Minerva Math [12], and OlympiadBench [5].

**Models.** We we select models range from 0.5B-3B [25]. We use 0.5B-ins and 1.5B-ins models as [32] found small base model may not follow the format prompt well. We also use Qwen2.5-Math-1.5B [26] to show the influence of math knowledge. For 3B model, we use Qwen-2.5-3B [25] base model. For all models, we conduct reinforcement fine-tuning using GRPO and NTHR using the same hyperparameters.

**Hyperparamters.** For the 0.5B model, we use two A6000 GPUs with a batch size of 32, maximum rollout length of 2500 tokens, learning rate $5e^{-7}$ and a mini-batch size of 16, resulting in two iteration updates per training step. We use math dataset to train the model for 105 steps.
For the larger models, we utilize four A100 GPUs with a batch size of 256, learning rate $1e-6$ and a mini-batch size of 64, leading to four iteration updates per step, for the math dataset, we train for 40 steps, which approximates one epoch. For the DeepScaler dataset, which contains more data, we train for 45 steps.
Across all models, we generate 8 rollouts per prompt. We use a default sampling temperature of 1.0, a clipping ratio of 0.2, and set the KL loss coefficient to $1 \times 10^{-4}$. The Qwen-math model [26] uses its full context length of 3072 tokens for rollouts, while all other models use a maximum rollout length of 4000 tokens. To accelerate training, we employ **dynamic sampling** [29], which filters out samples with zero advantage.
Lastly, we use $\beta = 1$ and define the weighting factor as $\eta = 2 \times |0.5 - p|$, where $p$ is the success rate. This formulation penalizes questions with success rates near 0.5 less, avoiding excessive reward for easy samples while not over-relying on difficult questions, whose scarce correct responses may

not yield effective token selection. Finally, we set the temperature to 0 to perform greedy decoding, following standard practice in math evaluation tasks [28, 2, 13]. This enables the assessment of the model's most confident output, which is essential for math tasks requiring exact correctness and providing users with consistent answers to repeated queries.

## 7.4 Complexity Discussion

In this section, we introduce several techniques to enhance the computational efficiency of NTHR, which is essential for practical implementation.

**Output last-layer embeddings with old policy probabilities.** Since GRPO relies on the old policy $\pi_{\texttt{old}}$ to compute the old probabilities used in the $\text{clip}(\cdot)$ operation, we only need to extract the last-layer embeddings during the forward pass of the old policy. This approach introduces no additional forward passes and thus keeps the computational overhead minimal. Moreover, last-layer embeddings have been widely adopted in prior work [33, 15] for constructing robust reward signals.

**Calculate summations first.** We more conveniently rewrite (7) as a matrix inner product.

$$\left\langle \sum_{i=1}^{N^+} \sum_{k=1}^{|\boldsymbol{y}_i^+|} \left( \mathbf{e}_{y_{i,k}^+} - \pi_{\theta(t)}(\cdot|\boldsymbol{x}, \boldsymbol{y}_{i,<k}^+) \right) \mathbf{h}_{\mathbf{x},\boldsymbol{y}_{i,<k}^+}^T, \left( \mathbf{e}_{\boldsymbol{y}_{j,k'}^-} - \pi_{\theta(t)}(\cdot|\boldsymbol{x}, \boldsymbol{y}_{j,<k'}^-) \right) \mathbf{h}_{\boldsymbol{x},\boldsymbol{y}_{j,<k'}^-}^T \right\rangle$$

Importantly, our reformulation involves calculating the summations over $i, k$ first before taking the inner product. This reduction reduces the overall complexity for compute (7) by 3.

**Focus on vocabulary of the responses.** As the formulation involves computing the outer product between the prediction error vector (e.g., $\mathbf{e}_{y_{i,k}^+} - \pi_{\theta(t)}(\cdot|\mathbf{x}, \mathbf{y}_{i,<k}^+)$) and the hidden embedding, which incurs a computational complexity of $O(|\mathcal{V}|d)$. Since the probability mass is primarily concentrated on the output words, for each question $\boldsymbol{x}$, we restrict the computation to the vocabulary $\mathcal{V}_{\boldsymbol{x}}^\star$ associated with its generated responses. Since $|\mathcal{V}_{\boldsymbol{x}}^\star| \ll |\mathcal{V}|$, this significantly lowers the overall cost to $O(|\mathcal{V}_{\boldsymbol{x}}^\star|d)$.

**Running time of each module.** We also track the average time cost of each module during training, as reported in Tab. 3. Notably, the data generation (Data Gen) module that using dynamic sampling accounts for the majority of the total training time. In contrast, the overhead introduced by NTHR is minimal, contributing only a small fraction to the overall cost. Notably, although deepseek-1.5B has a longer average output length of approximately 3,400 tokens, our NTHR still maintains a low time overhead.

| Model+dataset | Data Gen | Model Upd | THR | Ref | Old Prob | Total (Sec) |
|---|---|---|---|---|---|---|
| Qwen2.5-Math-1.5B (Math) | 250 | 140 | 22 | 55 | 55 | 522 |

Table 3: Average running time (per step, in seconds) of each module for different models and tasks.

$$\alpha_{k,k'} = \left\langle \mathbf{e}_{\boldsymbol{y}_{i,k}^+} - \pi_\theta(\cdot \mid \boldsymbol{x}, \boldsymbol{y}_{i,<k}^+), \mathbf{e}_{\boldsymbol{y}_{k'}} - \pi_\theta(\cdot \mid \boldsymbol{x}, \boldsymbol{y}_{<k'}) \right\rangle \tag{22}$$

As a result, our method does not introduce additional inference overhead, and the overall complexity remains small.

## 7.5 Performance across training iterations

In this section, we present the performance of various models across training iterations. As illustrated in Fig. 5, although performance fluctuates throughout training, NTHR consistently outperforms GRPO. Notably, for the Qwen-2.5-1.5B-Ins model, we report results at update step 100 (corresponding to 25 training steps), as performance begins to decline beyond this point.

## 7.6 Ablation study

In this section, we conduct ablation studies to shed more light on the role played by different modules.
**Ablation on $\beta$.** We conduct ablation study on $\beta$, i.e., the scaling factor for the threshold $\tau$ in Eq. (8), which is then used to control how bad the negative tokens should be masked out. We set $\beta$ with $\{-\infty, 0, 0.1, 1, \}$ and train with Qwen2.5-Math-1.5B using Math dataset. Choosing $\beta = -\infty$ corresponds to apply $\eta$ to all incorrect tokens. As shown in Tab. 4, this removal results in worse

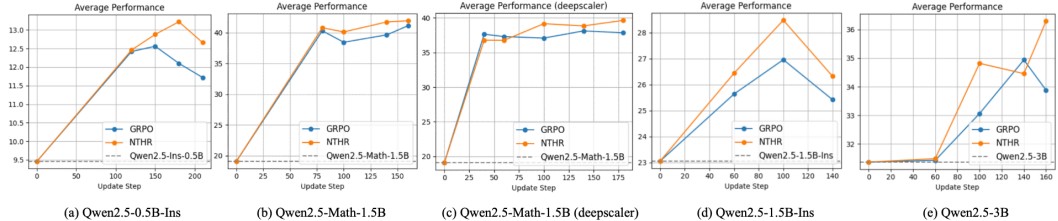

| (a) Qwen2.5-0.5B-Ins | (b) Qwen2.5-Math-1.5B | (c) Qwen2.5-Math-1.5B (deepscaler) | (d) Qwen2.5-1.5B-Ins | (e) Qwen2.5-3B |

Figure 5: Performance across training iterations for various models, NTHR consistently outperforms GRPO for most of the training process.

performance compared to GRPO. As further shown in Tab. 4, incorporating NTHR consistently improves performance over GRPO. While different $\beta$ values result in varying degrees of improvement, all settings outperform GRPO, indicating that reducing the influence of identified negative gradients is beneficial.

| Base model + Method | AIME24 | AMC | MATH500 | Minerva | Olympiad | Avg. |
|---|---|---|---|---|---|---|
| GRPO | 13.3 | 57.5 | **71.8** | 29.0 | 34.1 | 41.14 |
| NTHR ($\beta = -\infty$) | 13.3 | 50.0 | 71.8 | 30.5 | 34.4 | 40.00 |
| NTHR ($\beta = 0$) | **20.0** | 55.0 | 70.0 | 29.8 | 32.9 | 41.54 |
| NTHR ($\beta = 0.1$) | 13.3 | 57.5 | 71.4 | **30.9** | **34.4** | 41.50 |
| NTHR ($\beta = 1.0$) | 16.7 | **57.5** | 70.8 | 30.5 | 34.2 | **41.94** |

Table 4: Ablation study on $\beta$. Although changes in $\beta$ (except $\infty$) lead to variations in performance, they consistently improve GRPO, indicating that reduce influence of identified negative gradients is beneficial.

**Ablation on $\eta$.** We conduct an ablation study on $\eta$ using three variants: $\eta = 1 - p$, $\eta = p$, and $\eta = 2 \cdot |0.5 - p|$. As shown in Tab. 5, while the optimal choice of $\eta$ varies across different models, the variant $\eta = 2 \cdot |0.5 - p|$ consistently yields strong performance across all models. Therefore, we adopt $\eta = 2 \cdot |0.5 - p|$ in our training.

| Base model + Method | AIME24 | AMC | MATH500 | Minerva | Olympiad | Avg. |
|---|---|---|---|---|---|---|
| **Qwen2.5Math-1.5B** | | | | | | |
| GRPO | 13.3 | 57.5 | 71.8 | 29.0 | 34.1 | 41.14 |
| NTHR ($\eta = p$) | 13.3 | 55.0 | 72.4 | 29.0 | 33.3 | 40.60 |
| NTHR ($\eta = 1 - p$) | 13.3 | 62.5 | 72.8 | 30.5 | 34.2 | 42.66 |
| NTHR ($\eta = 2 \cdot |0.5 - p|$) | 16.7 | 57.5 | 70.8 | 30.5 | 34.2 | 41.94 |
| **Qwen2.5-3B** | | | | | | |
| GRPO | 6.7 | 35.0 | 66.6 | 31.2 | 29.9 | 33.88 |
| NTHR ($\eta = p$) | 10.0 | 47.5 | 64.6 | 33.8 | 26.8 | 36.54 |
| NTHR ($\eta = 1 - p$) | 6.7 | 42.5 | 63.8 | 32.4 | 28.0 | 34.68 |
| NTHR ($\eta = 2 \cdot |0.5 - p|$) | 10.0 | 47.5 | 65.6 | 31.6 | 26.8 | 36.30 |

Table 5: Ablation study on $\eta$.

## 7.7 Results on Llama

As shown in Fig. 6, the response length of Llama3.2-1B-Instruct declines rapidly after a few epochs, with the average length dropping from about 1.2K tokens to roughly 500. This reduction may stem from the model's limited cognitive behaviors [3], but once equipped with them, they can match Qwen's trajectory of self-improvement [3]. Despite this, We still test NTHR using Llama3.2-1B-Instruct and observed consistent improvements when applying our NTHR method as shown in Tab. 6.

## 7.8 NLL Loss as a Complementary

Recently, Yue et al. [31] proposed directly applying an NLL loss to positive samples, which aligns closely with the

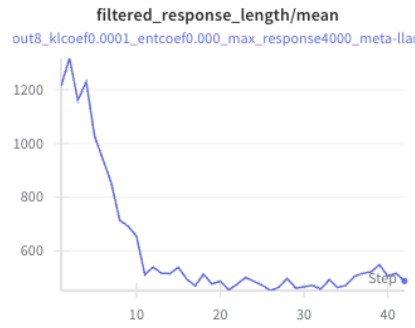

Figure 6: Response length dynamics of Llama3.2-1B-Instruct across different stages of GRPO training.

| Method | math500 | minerva_math | olympiad | aime24 | amc23 | avg |
|---|---|---|---|---|---|---|
| GRPO | 37.0 | 7.4 | 9.3 | 3.3 | 20.0 | 15.4 |
| + NLL Loss | 36.0 | **9.6** | **11.0** | 3.3 | **25.0** | **16.98** |
| + NTHR | **37.4** | 8.1 | **11.0** | 3.3 | 20.0 | 15.96 |

Table 6: Performance comparison across math reasoning benchmarks on Llama. Bold numbers indicate the best performance per column.

LLD issue we identify, as it increases the likelihood of correct responses. Specifically, incorporating a supervised NLL term reinforces the model's confidence in generating correct outputs. Motivated by this, we augmented GRPO with an additional NLL loss and trained it using Llama-3.2-1B-Instruct. As shown in Tab. 6, this modification leads to improved performance across multiple tasks. The NLL objective tackles LLD from a complementary perspective—strengthening correct responses through positive gradients—while NTHR mitigates harmful negative updates. Both methods consistently yield performance gains.

## 7.9 Results on Deepseek-1.5B

We utilize four A100 GPUs with a batch size of 256 and a mini-batch size of 64, resulting in four update iterations per training step. Each prompt generates 8 rollouts. We adopt a default sampling temperature of 0.6, set the clipping ratio to 0.2, and use a KL loss coefficient of $1 \times 10^{-4}$. The maximum rollout length is capped at 4000 tokens. We set $\beta = 0$ and define the weighting factor as $\eta = 2 \cdot |0.5 - p|$. We limit training to 35 steps, as we observed a consistent decline in response length with continued training.

Since the context window length is limited, the results are very unstable, as a results, we calculate the last two checkpoints' average performance and report the results in the table below. Although this setting may not be optimal or robust, we include the results and observe that our NTHR consistently outperforms GRPO on average.

| Base model + Method | AIME24 | AMC | MATH500 | Minerva | Olympiad | Avg. |
|---|---|---|---|---|---|---|
| **Deepseek-1.5B** | | | | | | |
| Base | 20.0 | 50.0 | 68.8 | 20.6 | 29.6 | 37.8 |
| GRPO | 25.0 | 65.0 | **81.10** | **34.6** | 40.0 | 49.1 |
| NTHR | 25.0 | **70.0** | 80.60 | 33.7 | **40.7** | **50.0** |

Table 7: Results across selected math benchmarks for deepseek-1.5B.

## 7.10 Examples of LLD Incorrect Responses

We present the top-ranked incorrect response based on LLD for Qwen-2.5Math-1.5B in Fig. 1c. As shown in Tab. 8, the answer is nearly correct, with the only issue occurring in the final step.

Similarly, Fig. 1b shows the top-ranked incorrect response by LLD for Deepseek-1.5B. As detailed in Tab. 9, the response includes the correct answer multiple times; however, due to continued self-reflection, it becomes truncated by the context window limit, resulting in a final answer that is wrong.

## 7.11 Words with high NTHR

We present additional examples of words with high NTHR scores. As shown in Tab. 10, using the same response from Tab. 8, we observe that the correct parts are associated with high NTHR values, while the incorrect portion in step 5 receives low NTHR values (in black). We also show response with incorrect output format in Tab. 11, using the same response from Fig. 2, we observe that core steps and correct answer 8 are with high NTHR.

| Model & Data | Problem & Answer | Incorrect Response |
|---|---|---|
| Qwen2.5math-1.5B
Math (order 1) | **Question**: A portion of the graph of a quadratic function $f(x)$ is shown.

Let $g(x) = -f(x)$, $h(x) = f(-x)$. If $a$ is the number of points where the graphs of $y = f(x)$ and $y = g(x)$ intersect, and $b$ is the number of points where the graphs of $y = f(x)$ and $y = h(x)$ intersect, what is $10a + b$?
**Answer**: 21 | To solve the problem, we need to analyze the given functions $f(x)$, $g(x)$, and $h(x)$ and determine the points of intersection between the graphs of these functions.
**Graph of $f(x)$:**
The graph of $f(x)$ is a quadratic function. From the given graph, we can observe the following:

• The roots of $f(x)$ are at $x = 1$ and $x = 3$.

• The vertex of the parabola is at $x = 2$.

**Graph of $g(x) = -f(x)$:**
The graph of $g(x)$ is the reflection of the graph of $f(x)$ across the x-axis. Therefore:

• The roots of $g(x)$ are also at $x = 1$ and $x = 3$.

• The vertex of the parabola for $g(x)$ is at $x = 2$.

**Graph of $h(x) = f(-x)$:**
The graph of $h(x)$ is the reflection of the graph of $f(x)$ across the y-axis. Therefore:

• The roots of $h(x)$ are at $x = -1$ and $x = -3$.

• The vertex of the parabola for $h(x)$ is at $x = -2$.

Next, we determine the points of intersection:
**Intersection of $f(x)$ and $g(x)$:**
Since $g(x) = -f(x)$, the graphs of $f(x)$ and $g(x)$ intersect at the points where $f(x) = -f(x)$. This simplifies to:
$$f(x) = 0$$
The roots of $f(x)$ are at $x = 1$ and $x = 3$. Therefore, the graphs of $f(x)$ and $g(x)$ intersect at two points:
$$a = 2$$
**Intersection of $f(x)$ and $h(x)$:**
Since $h(x) = f(-x)$, the graphs of $f(x)$ and $h(x)$ intersect at the points where $f(x) = f(-x)$. This implies that $f(x)$ is an even function, and its graph is symmetric about the y-axis. The roots of $f(x)$ are at $x = 1$ and $x = 3$, but these are not symmetric about the y-axis. Therefore, the graphs of $f(x)$ and $h(x)$ do not intersect. Thus: $b = 0$
Finally, we calculate $10a + b$: $10a + b = 10 \cdot 2 + 0 = 20$
Thus, the final answer is:
$$\boxed{20}$$ |

Table 8: Incorrect response example from `qwen2.5math-1.5B`. The incorrect response is almost correct.

| Model & Data | Problem & Answer | Incorrect Response |
|---|---|---|
| Deepseek-1.5B AIME (order 1) | **Question**: One commercially available ten-button lock may be opened by pressing – in any order – the correct five buttons. The sample shown below has $\{1, 2, 3, 6, 9\}$ as its combination. Suppose that these locks are redesigned so that sets of as many as nine buttons or as few as one button could serve as combinations. How many additional combinations would this allow? **Answer**: 770 | To solve the problem, explain your reasoning step by step, and put your final answer within boxed. Alright, so I have this problem about ten-button locks and figuring out how many additional combinations a redesign would allow. Let me try to break it down step by step. First, the original lock. It seems like a ten-button lock where you need to press five buttons in any order to open it. The combination given is 1,2,3,6,9, which is a set of five buttons. So, the question is asking about how many additional combinations would be possible if we redesign the locks to allow sets of as many as nine buttons or as few as one button. I think I need to figure out the total number of possible combinations before and after the redesign and then find the difference. That should give me the number of additional combinations. Starting with the original lock |
| | | **...** |
| | | Therefore, the number of additional combinations is $1022 - 252 = 770$.Wait, thatś the same answer as before. So, perhaps 770 is the correct answer. But wait, let me make sure. Is the original number of combinations one, or 252? |
| | | **...** |
| | | Then, the redesigned lock allows for any number of buttons from one to nine, so $2^10 - 2 = 1022$ possible combinations, but the original combination was just one. So, the number of additional combinations would be $1022 - 1 = 1021$. But that seems like a lot more. Wait, which is it? Is the original combination one specific combination, or is it 252 unique combinations? In the problem, it says, "one commercially available ten-button lock may be opened by pressing – in any order – the correct five buttons." |
| | | **...** |
| | | After redesign, the number of possible combinations is 1022. So, the number of additional combinations is $1022-1 = 1021$. But wait, the redesign allows for any combination from one to nine buttons. So, the number of possible combinations is 1022, which includes the original 252 combinations. Therefore, the number of additional combinations is $1022 - 252 = 770$. |
| | | **...** |
| | | The redesigned allows for more combinations, but if you consider that each combination is a set of buttons, then the number of additional combinations is $1022 - 252 = 770$. But the problem says, "the sample shown below has 1,2,3,6,9 as its combination." So, the sample is a specific combination, so thatś one combination. So, originally, the number of possible combinations was 252, including the correct one. So, if you redesign it, the number of possible combinations is 1022, which includes all the original 252 combinations plus $1022 - 252 = 770$ additional combinations. Therefore, the number of additional combinations is 770. But to make sure, let me think of another approach. If the original lock had only one combination, the number of possible combinations is 252, each being a unique set of five buttons. Then, if you redesign it so that you can have any number of buttons from one to nine, the number of possible combinations is Stop due to limited context window length. |

Table 9: Incorrect response example from `deepseek-1.5B` on AIME: Although the response includes the correct answers, excessive self-reflection causes the output to exceed the context window, preventing it from being formatted correctly. Green dashes means omitted reasoning.

| Model & Data | Problem & Answer | Incorrect Response |
|---|---|---|
| Qwen2.5math-1.5B
Math | **Question**: A portion of the graph of a quadratic function $f(x)$ is shown.

Let $g(x) = -f(x)$, $h(x) = f(-x)$. If $a$ is the number of points where the graphs of $y = f(x)$ and $y = g(x)$ intersect, and $b$ is the number of points where the graphs of $y = f(x)$ and $y = h(x)$ intersect, what is $10a + b$?
**Answer**: 21 | To solve the problem, we need to analyze the given functions $f(x)$, $g(x)$, and $h(x)$ and determine the points of intersection between the graphs of these functions.
1. Graph of $f(x)$: The graph of $f(x)$ is a quadratic function. From the given graph, we can observe the following: - The roots of $f(x)$ are at $x = 1$ and $x = 3$. - The vertex of the parabola is at $x = 2$.
2. Graph of $g(x) = -f(x)$:The graph of $g(x)$ is the reflection of the graph of $f(x)$ across the x-axis. Therefore: - The roots of $g(x)$ are also at $x = 1$ and $x = 3$. - The vertex of the parabola for $g(x)$ is at $x = 2$.
3.Graph of $h(x) = f(-x)$:** The graph of $h(x)$ is the reflection of the graph of $f(x)$ across the y-axis. Therefore: - The roots of $h(x)$ are at $x = -1$ and $x = -3$. - The vertex of the parabola for $h(x)$ is at $x = -2$.
Next, we determine the points of intersection:
4. Intersection of $f(x)$ and $g(x)$: Since $g(x) = -f(x)$, the graphs of $f(x)$ and $g(x)$ intersect at the points where $f(x) = -f(x)$. This simplifies to:

$$f(x) = 0$$

The roots of $f(x)$ are at $x = 1$ and $x = 3$. Therefore, the graphs of $f(x)$ and $g(x)$ intersect at two points:

$$a = 2$$

5. Intersection of $f(x)$ and $h(x)$: Since $h(x) = f(-x)$, the graphs of $f(x)$ and $h(x)$ intersect at the points where $f(x) = f(-x)$. This implies that $f(x)$ is an even function, and its graph is symmetric about the y-axis. The roots of $f(x)$ are at $x = 1$ and $x = 3$, but these are not symmetric about the y-axis. Therefore, the graphs of $f(x)$ and $h(x)$ do not intersect.
Thus:
$$b = 0$$
Finally, we calculate $10a + b$:

$$10a + b = 10 \cdot 2 + 0 = 20$$

Thus, the final answer is:

$$\boxed{20}$$ |

Table 10: High NTHR (highlight in red) example from `qwen2.5math-1.5B`. Core steps in correct parts are with high NTHR. The erroneous words of step 5 are with low NTHR values.

| Model & Data | Problem & Answer | Incorrect Response |
|---|---|---|
| Qwen2.5-0.5B-Ins Math | **Question**: What is the greatest integer $n$ such that $n^2 - 11n + 24 \leq 0$? 
 **Answer**: 8 | To solve the inequality $n^2 - 11n + 24 \leq 0$, we need to find the values of n that satisfy the inequality. Since the inequality is an inequality, we should first find the roots of the corresponding equation $n^2 - 11n + 24 = 0$. Let's solve the equation $n^2 - 11n + 24 = 0$: 1. Factor the quadratic equation: $n^2 - 11n + 24 = (n-3)(n-8) = 0$. 2. Set each factor equal to zero and solve for n: $n - 3 = 0$ or $n - 8 = 0$. 3. Solve for n: $n = 3$ or $n = 8$. Therefore, the roots of the equation $n^2 - 11n + 24 = 0$ are $n = 3$ and $n = 8$. Now, we need to find the values of n that satisfy the inequality $n^2 - 11n + 24 \leq 0$. Since the inequality is an inequality, we should check the intervals between the roots. The roots $n = 3$ and $n = 8$ divide the number line into three intervals: $(-\infty, 3]$, $[3, 8]$, and $[8, \infty)$. Let's test a point in each interval to determine where the inequality holds true. 1. For n in $(-\infty, 3)$, let's test $n = 0$: $0^2 - 11(0) + 24 = 24 > 0$. 2. For n in $(3, 8)$, let's test $n = 5$: $5^2 - 11(5) + 24 = 25 - 55 + 24 = -4 < 0$. 3. For n in $(8, \infty)$, let's test $n = 9$: $9^2 - 11(9) + 24 = 81 - 99 + 24 = 6 > 0$. Therefore, the inequality $n^2 - 11n + 24 \leq 0$ holds true for $3 \leq n \leq 8$. Hence, the greatest integer $n$ that satisfies the inequality is $n = 8$. In boxed format, the answer is 8. |

Table 11: High NTHR (highlight in red) example from `qwen2.5-0.5B-Ins`. This example have correct answer but in wrong format. Core steps and correct answer $8$ are with high NTHR.

