# OpenReview forum: "On the Effect of Negative Gradient in Group Relative Deep Reinforcement Optimization"
_NeurIPS.cc/2025/Conference — NeurIPS 2025 poster_

### Official Review · Reviewer_rZbD · 2025-06-12

**Clarity:** 3
**Significance:** 2
**Originality:** 3
**Rating:** 3
**Confidence:** 3

**Summary:**

The paper identifies the Lazy Likelihood Displacement problem, where the probability of generating correct answers increases slowly or even decreases during RL fine-tuning for LLMs, and analyzed theoretically the origin of this problem.
The authors then proposed Negative Token Hidden Reward (NTHR) to account for how positive tokens and negative tokens are related, and perform more fine-grained control over the estimated advantage.
Empirical results indicate that the proposed method can improve LLM math reasoning.

**Questions:**

1. It seems to me that a symmetric argument can also be made about "positive gradients", can the authors explain why only negative gradients were considered?
2. Since the proposed fix is essentially biasing the policy gradient, can the authors elaborate on what are the implications (e.g., convergence, bias-variance tradeoff) of it in the context of RL?

**Ethical Concerns:**

["NO or VERY MINOR ethics concerns only"]

**Final Justification:**

My major concerns regarding the lack of theoretical justification and marginal gain of the empirical performance are not addressed. Consequently, I'm maintaining my original score.

**Limitations:**

yes

**Quality:**

2

**Strengths And Weaknesses:**

Strengths:

1. The paper is clearly written and easy to follow.
2. The qualitative results are compelling. In particular, the finding that NTHR tokens can be used to identify correct tokens in incorrect responses is quite interesting, and may open up new ways to design RLVR algorithms.


Weakness:

1. The theoretical motivation appears to rest on the assumption that "negative gradients decreasing the likelihood of correct response is inherently problematic"; however, from an RL perspective, that this is not necessarily the case.
Since we ultimately care about maximizing the expected return, and the policy gradient only guarantees that, locally, following the gradient can increase the expected return of the policy.
It is, therefore, acceptable that the gradient decreases the likelihood of positive-reward trajectories, provided this is outweighed by the reductions in the likelihood of negative-reward trajectories.
2.  Note that every time the policy changes, it can also discover new correct responses.
Consequently, it is not clear to me whether Eq (2) is a good metric, as it is only computed over previously collected responses.
3. The proof of Theorem 4.4 is problematic.
Assumption (line 750) should be put into theorem statements rather than the proof.
Additionally, it is not clear to me why (III, Eq 20) and (IV, Eq 21) can be omitted.
4. The empirical results show marginal improvements over GRPO. While small improvements are not necessarily a concern, the lack of error bars makes it difficult to assess the reliability of the reported gains.

---

> ### Author Response · Authors · 2025-07-31
> **answer to weakness and questions**
>
> --------- Weakness 1& Q2: theory analysis --------
>
> **Answer**:  We appreciate the reviewer’s perspective and agree that, from a standard RL standpoint, convergence guarantees and theoretical properties like bias-variance tradeoff are central concerns. However, in practice—especially in the context of LLMs—many classical RL assumptions do not hold. For example, if we treat the context as the state and the next token as the action, the action space becomes exponentially large `(∼|V|^L)`, and collecting representative rollouts becomes infeasible, as typical setups involve fewer than 100 rollouts per prompt.
> Due to these limitations, most RL methods for LLMs operate under approximations that differ significantly from the theoretical RL setting. That’s why our approach is more pragmatic: we start by identifying undesirable training behaviors such as Lazy Likelihood Displacement (LLD), and then introduce targeted modifications—like selective gradient downweighting—that lead to consistent performance improvements.
>
> ----------- Weakness 2:  new correct responses -------
>
> **Answer**: We would like to clarify that Equation (2) is indeed computed in an **online manner**. Specifically, since GRPO operates as an online learning algorithm, new responses are continually sampled from the evolving policy. This means the set of “newly correct responses” naturally emerges during training and is directly incorporated into the LLD measurement. As a result, LLD reflects the log-likelihood dynamics on **freshly sampled correct responses from the current policy**.
>
> ----------- Weakness 3:  question on Theorem 4.4  -------
>
> **Answer**:  We appreciate your careful reading and thoughtful feedback. We agree that the assumption in line 750 should be part of the theorem statement and will revise Theorem 4.4 accordingly.
>
> We did not omit token unembeddings, as explicitly acknowledged in Theorem 4.4 by the phrase “in addition to the dependence on token unembeddings.”  Our focus on terms (I) and (II) is based on three key considerations: (1) token embeddings reflect the influence of all network parameters except the unembedding layer; (2) they are shaped by the specific words in each sample, covering a broader representational space than token unembeddings; and (3) empirical evidence in Table 1 shows that the difference (II) − (I) consistently correlates with likelihood displacement. For these reasons, Theorem 4.4 centers on the dominant contributions from (I) and (II).
>
> ----------- Q 1:  positive gradients -------
>
> **Answer**:  Thank you for this insightful question. While a symmetric analysis of positive gradients is certainly possible, our primary focus is on negative gradients, as they are the key contributors to the undesirable behavior we identify as Lazy Likelihood Displacement (LLD). Specifically, we find that negative gradients can inadvertently reduce the likelihood of correct responses, particularly when correct and incorrect samples share overlapping features.
> That said, we agree that the role of positive gradients is also important and worth further exploration. Motivated by your suggestion, we conducted additional experiments on Qwen-1.5B-Math, a model with strong mathematical reasoning capabilities. We extended our method by not only downweighting selected negative gradients but also amplifying influential positive gradients. We denote this combined strategy as THR (Token Hidden Reward). As shown in the results below, THR yields further improvements in greedy decoding performance across several math benchmarks, outperforming both GRPO and NTHR:
>
> | Method | math500 | minerva_math | olympiad | aime24 | amc23 | avg   |
> |--------|---------|--------------|----------|--------|-------|--------|
> | GRPO   | **71.8** | 29.0         | 34.1     | 13.3   | 57.5  | 41.14  |
> | NTHR   | 70.8     | 30.5         | 34.2     | **16.7** | 57.5  | 41.94  |
> | THR    | 71.4     | **33.1**     | **34.5** | 13.3   | **62.5** | **43.00** |
>
> These results suggest that positive gradient enhancement can be a promising complementary direction to our current approach. We view this as an exciting avenue for future research.

---

> > ### Comment · Reviewer_rZbD · 2025-08-05
> >
> > I thank the authors for the detailed reply.
> >
> > > [...] Due to these limitations, most RL methods for LLMs operate under approximations that differ significantly from the theoretical RL setting [...]
> >
> > Can the authors elaborate on this? To the best of my knowledge, popular algorithms like GRPO or PPO do not differ significantly from their theoretical underpinnings. In fact, methods guided by theoretical considerations have shown to improve RL training for LLMs (e.g., [1]).
> >
> > > We would like to clarify that Equation (2) is indeed computed in an online manner. Specifically, since GRPO operates as an online learning algorithm, new responses are continually sampled from the evolving policy. This means the set of “newly correct responses” naturally emerges during training and is directly incorporated into the LLD measurement. As a result, LLD reflects the log-likelihood dynamics on freshly sampled correct responses from the current policy.
> >
> > Sorry if I'm not stating the problem clearly, but my main concern is that it is not enough to consider $(x,y)$ pairs generated  from the model. This is because you'll only see high probability pairs from the generated responses, and the measure (Eq 2) will not correctly reflect the changes in rare positive pairs. In fact, from this perspective, it is not too surprising that LLD arises, because the metric is computed over samples with high probabilities that are already saturating.
> >
> > [1] Liu, Zichen, et al. "Understanding r1-zero-like training: A critical perspective." arXiv preprint arXiv:2503.20783 (2025).

---

> ### Author Response · Authors · 2025-08-04
> **Reaching the End of the Public Discussion Phase**
>
> Dear Reviewer,
>
> Thank you very much for your efforts in the reviewing process, we have added experiments and analysis to answer your questions and hope that our responses have sufficiently addressed the concerns you raised. We welcome further discussion if you have more questions and suggestions.
>
> As the discussion deadline is approaching, we would be very grateful if you could take a moment to review our reply. Thank you for your time and consideration.
>
> Best Authors

---

> ### Author Response · Authors · 2025-08-06
>
> Thank you very much for your feedback and thoughtful comments!
>
> ------ Q1: elaborate on theory analysis-------
>
> **Answer**:  We thank the reviewer for the clarification and agree that PPO/GRPO are grounded in RL theory. Our point was not to dispute this, but to emphasize that in LLM settings, modifications to the loss (such as NTHR) can meaningfully affect training dynamics in ways not fully captured by classical theory, particularly regarding how gradient signals influence the likelihood of correct responses.
>
> To do this, we resort to more of what we would characterize as “behavioral analysis” (i.e. modeling and analyzing the empirical behavior of the system under a given RL objective) which is complementary to classical RL theory and can make strides to understanding complex systems like RL-LLMs.
>
> There are several examples that highlight the need for such a complementary approach. Many recent versions of GRPO make substantial changes to the original GRPO loss, not just in papers (which often provide some theoretical justification), but also in many open-source implementations. For example, removing the KL term [1], raising the positive clip threshold [1], adding NLL loss on correct responses [2], or even dropping one response side altogether [3].
>
> While classical RL intuition might suggest that these modifications could break training (e.g., by removing the KL term) or introduce bias (e.g., through increased clipping or added NLL loss), in LLM settings they often still work, and in some cases, work remarkably well, even when the original assumptions no longer hold.
>
> [1] DAPO: An Open-Source LLM Reinforcement Learning System at Scale
>
> [2] VAPO: Efficient and Reliable Reinforcement Learning for Advanced Reasoning Tasks.
>
> [3] REINFORCE++: An Efficient RLHF Algorithm with Robustness to Both Prompt and Reward Models
>
> ------ Q2: elaborate on LLD and rare responses-------
>
> '' This is because you'll only see high probability pairs from the generated responses, and the measure (Eq 2) will not correctly reflect the changes in rare positive pairs. ''
>
> **Answer**: We hope the following remarks clarify with Eq. (2) is useful:
>
> 1. **Sampled responses are critical to monitor, as they drive training** : It is the sampled responses, not unsampled rare ones, that directly shape the gradient updates. If GRPO fails to increase the likelihood of these observed correct responses, or worse, decreases it, it indicates a fundamental misalignment. LLD is specifically intended to detect and mitigate this issue.
>
> 2. **Rare responses can still suffer from LLD if sampled**: Our theoretical analysis (e.g., Theorem 4.4 negative gradient term) shows that even rare correct responses can be affected by LLD when similar incorrect counterparts exist. Due to the model’s uncertainty on rare responses, such incorrect variants are more likely to be sampled and penalized, thereby indirectly suppressing the correct response. If a rare correct response is sampled and suffers from LLD, our method can address it.
>
> 3. **High sampling temperature promotes diversity**: We use a sampling temperature of 1, which encourages diversity and enables the model to generate low-likelihood correct responses. As a result, LLD captures training dynamics across a wide range of response probabilities.
>
> —------- In fact, from this perspective, it is not too surprising that LLD arises, because the metric is computed over samples with high probabilities that are already saturating. —------
>
> **Answer**:  We respectfully disagree with the claim that LLD is unsurprising simply because the sampled responses are already saturated.
>
> 1. **Responses are not saturated**: These responses are not yet updated when measured. As stated in line 121, we reinitialize the model parameters for each individual sample, meaning each sample is trained on an unsaturated pretrained model. Since our measure is at the beginning of training, their confidence is not saturated.
>
> 2. **Empirical evidence contradicts the saturation hypothesis**: In Figure 1, we observe that removing the influence of the negative response leads to a significantly greater increase in the confidence of the positive responses, indicating they were not saturated. Furthermore, the confidence of some positive responses even decreases after a single GRPO step, highlighting the severity of the LLD effect.
>
> Thank you again for the thoughtful questions. We will clarify these points in the revision as discussed above.

---

> > ### Comment · Reviewer_rZbD · 2025-08-07
> >
> > I thank the authors for the reply. However, I share the concerns raised by Reviewer RWsJ and Reviewer Nhas regarding the lack of theoretical grounding and the limited empirical evidence. In my view, these issues make it difficult to justify the claimed link between LLD and performance. Consequently, I maintain my original score.

---

> ### Author Response · Authors · 2025-08-07
>
> Thank you for your engagement and thoughtful review.
>
> We respectfully disagree with the assessment that the paper lacks sufficient empirical evidence. Our submission includes extensive experiments across multiple mathematical reasoning benchmarks (e.g., Math500, AIME, AMC), consistently showing performance improvements when LLD is mitigated via our proposed method. In response to reviewer suggestions, we have also added new experiments on the MedCQA dataset (medical QA), Qwen-Coder on the LeetCode dataset (code generation), which confirm the presence of LLD in these domains, and results on LLaMA-2 models to demonstrate the effectiveness of our approach across different model families. These additions will be included in the revised version.
>
> We also maintain our position that our theoretical analysis, despite the complexity of the RL setting, offers non-trivial and practically relevant conclusions. In particular, Theorem 4.4 formally characterizes how token-level interference leads to LLD, and our empirical findings support the predicted effects on model behavior. We believe this represents a meaningful step toward better understanding and improving RL-based training for LLMs in real-world applications.
>
> We appreciate your continued engagement and hope the additional results and clarifications help address your concerns.

---

### Official Review · Reviewer_RWsJ · 2025-06-27

**Clarity:** 3
**Significance:** 3
**Originality:** 3
**Rating:** 4
**Confidence:** 3

**Summary:**

This paper investigates an problem in Group Relative Policy Optimization (GRPO) termed Lazy Likelihood Displacement (LLD), where the likelihood of correct responses sees marginal or even negative updates during training. To address this problem, this paper proposes a novel method called Negative Token Hidden Reward (NTHR), which selectively reduces the penalty on tokens within incorrect responses that are identified as contributing to LLD.

**Questions:**

Both theoretical analysis and algorithm design involve unconstrained embedding $h$. However, it's unclear to me what the author defines as unconstrained embedding $h$ in the context of the code. In other words, how does a neural network model translate into unconstrained embedding $h$ in your code?

**Ethical Concerns:**

["NO or VERY MINOR ethics concerns only"]

**Limitations:**

See weakness and questions.

**Quality:**

3

**Strengths And Weaknesses:**

Strengths:

The theoretical analysis in this paper is robust, although Assumption 4.3 might be somewhat oversimplified.

Weaknesses:

- The experimental analysis lacks comprehensiveness, as the paper only tests on mathematical datasets.

- Although LLD appears intuitively correct that it might lead to decreased performance. Nevertheless, the paper neither provides theoretical analysis nor experimental examples to support this claim.

---

> ### Author Response · Authors · 2025-07-31
> **answer to weakness and questions**
>
> -------- Weaknesses 1:  more tasks --------
>
> **Answer**: Thank you for the comment. We acknowledge the importance of broader evaluation. We chose mathematical reasoning as our primary domain because it is widely used as a benchmark for reinforcement learning in LLMs (e.g., DeepSeek-Math, QwenMath).
> Following your suggestion, we also tested on the MedCQA medical dataset. Using the first 50 questions, we observed a clear LLD effect: the log-likelihood change ranged from 0.4 (min) to 2.73 (max), consistent with the behavior seen in math tasks. This provides early evidence that LLD is not confined to a single domain. We plan to further extend our experiments to additional settings, including code generation, in future work. We will however include the new experiment on the MedCQA medical dataset in the revision.
>
> -------- Weaknesses 2:  theory and empirical on LLD --------
>
> **Answer**:  We respectfully disagree with the claim that our paper lacks both theoretical and empirical support for the impact of LLD. In fact, our analysis explicitly identifies the mechanism by which LLD arises—namely, through negative gradients affecting shared tokens between correct and incorrect responses. This is formalized in Theorem 4.4, where we decompose the gradient contributions and show how likelihood displacement is linked to token-level interactions.
> Empirically, we demonstrate that mitigating LLD via our NTHR method leads to consistent improvements in final task performance across diverse mathematical benchmarks (Table 1). These gains directly validate the practical consequences of LLD and show that addressing it improves model behavior. We also provide qualitative evidence in Figures 1 and 4, where reduced or negative log-likelihood shifts are shown to correlate with under-optimization of correct responses.
> Together, these theoretical and empirical results provide strong support for the significance of LLD.
>
> ------- Q1: unconstrained embedding in the context of the code -------
>
> **Answer** : Thank you for the question. In our work, “unconstrained embedding” specifically refers to the last-layer output embeddings, as stated in line 178.  Following your suggestion, we will further highlight this in our revised version.

---

> ### Author Response · Authors · 2025-08-04
> **Reaching the End of the Public Discussion Phase**
>
> Dear Reviewer,
>
> Thank you very much for your efforts in the reviewing process, we have added discussion and analysis to answer your questions and hope that our responses have sufficiently addressed the concerns you raised. We welcome further discussion if you have more questions and suggestions.
>
> As the discussion deadline is approaching, we would be very grateful if you could take a moment to review our reply. Thank you for your time and consideration.
>
> Best Authors

---

> > ### Comment · Reviewer_RWsJ · 2025-08-06
> >
> > Thanks for your response. I maintain my score. However, I am still not very satisfied with the number of types of tasks tested. I hope the author can add experiments on code tasks.

---

> > > ### Author Response · Authors · 2025-08-06
> > >
> > > Thank you again for your thoughtful feedback and for the positive score.
> > >
> > > In response to your suggestion, we conducted a preliminary study on LLD in the code generation domain using the Qwen-Coder-3B-Instruct model on the LeetCode dataset. We observed a minimum log-likelihood change of 0.036 and a maximum of 0.27, indicating that the LLD issue also arises in this domain. We hope this early result helps address your concern and further supports the generality of the LLD phenomenon beyond mathematical reasoning.
> > >
> > > We are currently working on code tasks. While we cannot guarantee that the results will be ready before the deadline due to resource limitations and time constraints, we will do our best to include them if possible.
> > >
> > > We truly appreciate your constructive input throughout the review process and thank you again for your time, consideration, and support.

---

### Official Review · Reviewer_Nhas · 2025-07-02

**Clarity:** 4
**Significance:** 2
**Originality:** 2
**Rating:** 4
**Confidence:** 4

**Summary:**

This paper investigates the impact of negative gradients on the likelihood of correct responses in GRPO. The authors identify a phenomenon they call Lazy Likelihood Displacement (LLD), where the penalization of incorrect responses can inadvertently reduce or lead to small likelihood changes of correct ones. To address this issue, they propose a method called Negative Token Hidden Reward, which selectively penalizes the advantage of tokens in incorrect responses that contribute most to lowering the likelihood of correct responses. The authors demonstrate the effectiveness of their approach while improving performance on math reasoning tasks across models of varying sizes (0.5B to 3B parameters). The key contributions of this paper are: (1) the identification of LLD in GRPO, (2) the development of their method to address LLD, and (3) the empirical validation of their method's effectiveness in improving performance on math reasoning tasks.

**Questions:**

Questions:
- 1) Can you guarantee convergence to an optimal policy with this new loss?
- 2) Can you relate this new advantage estimation to classical GAE?

**Ethical Concerns:**

["NO or VERY MINOR ethics concerns only"]

**Final Justification:**

My main concerns were addressed, so I've raised my score.

**Limitations:**

yes

**Quality:**

2

**Strengths And Weaknesses:**

Strenghts:
- The paper is well-structured and provides a clear, step-by-step explanation of the problem.
- The paper is sound, the theorem and the lemma are correct.
Weaknesses:
- The paper lacks some theoretical analysis of the impact of using the new loss (convergence to an optimal policy).
- The experiments are limited to Qwen2.5 on math reasonning. It has been shown that this model can exhibit unexpected behavior, especially in this setting, compared to others [1]. It would be better to confirm that the method also works with other models.
- The significance and novelty are a bit limited. The proposed method introduces a token-level advantage estimation coming from the LLM itself instead of an independent critic. Therefore, one would expect a comparison with a method using classical token-level advantage estimation like PPO.
- The method introduces 3 new hyperparameters to GRPO. I acknowledge that an ablation study of those is provided in the appendix. However, it is often easy to beat a method by introducing hyperparameters to it that are tuned to beat the base version on a specific problem.
[1] Shao, Rulin et al. “Spurious Rewards: Rethinking Training Signals in RLVR.” (2025).

---

> ### Author Response · Authors · 2025-07-31
> **answer to weakness and questions**
>
> -----------   Weaknesses 1 & Q1,2:  theoretical analysis --------
>
> **Answer**: We appreciate the reviewer’s perspective and agree that, from a standard RL standpoint, convergence guarantees and theoretical properties like bias-variance tradeoff are central concerns. However, in practice—especially in the context of LLMs—many classical RL assumptions do not hold. For example, if we treat the context as the state and the next token as the action, the action space becomes exponentially large `(∼|V|^L)`, and collecting representative rollouts becomes infeasible, as typical setups involve fewer than 100 rollouts per prompt.
> Due to these limitations, most RL methods for LLMs operate under approximations that differ significantly from the theoretical RL setting. That’s why our approach is more pragmatic: we start by identifying undesirable training behaviors such as Lazy Likelihood Displacement (LLD), and then introduce targeted modifications—like selective gradient downweighting—that lead to consistent performance improvements.
>
> ------------ Weaknesses 2:  unexpected behavior of qwen and add llama --------
>
> **Answer**: Thank you for the comment. While we acknowledge that Qwen2.5 has stronger mathematical capabilities—as noted in [1]—it’s important to highlight that even in that work, correct reward signals still lead to significantly larger gains than random or incorrect ones, reinforcing the validity of reward-based fine-tuning for this model family.
> Moreover, recent findings [2] suggest that Qwen’s advantage in reasoning tasks is not solely due to its knowledge, but also its cognitive behaviors—such as verification and backtracking—which it exhibits naturally. In contrast, models like LLaMA initially lack these behaviors, but once equipped with them, they can match Qwen’s trajectory of self-improvement.
> To address concerns about generality, we extended our experiments to LLaMA3-1B-Ins, and observed consistent improvements when applying our NTHR method. As shown below, the performance trends align closely with those observed on Qwen2.5, suggesting that our method generalizes well across model families.
>
> | Method         | math500 | minerva_math | olympiad | aime24 | amc23 | avg    |
> |----------------|---------|--------------|----------|--------|-------|--------|
> | GRPO    | 37.0    | 7.4          | 9.3      | 3.3    | 20.0  | 15.4   |
> | +NTHR     | **37.4**    | **8.1**          | **11.0**     | 3.3    | 20.0  | **15.96**  |
>
>
> [1] Spurious Rewards: Rethinking Training Signals in RLVR
>
> [2] Cognitive Behaviors that Enable Self-Improving Reasoners, or, Four Habits of Highly Effective STaRs
>
>
> --------- Weaknesses 3: NTHR token-level advantage estimation and  PPO  -----------
>
> **Answer**:  Thank you for the comment. We appreciate that you recognize the potential of our method as a model-free alternative to classical value-based methods like PPO. However, we want to emphasize that our primary contribution is not a new token-level advantage estimator,  but on diagnosing and mitigating LLD, a specific optimization pathology in GRPO.
>
> While PPO also operates at the token level, it relies on an external value network which fundamentally differs from the value-free setup of GRPO. Therefore, a direct comparison with PPO is not straightforward and would not isolate the LLD-specific behaviors we aim to study. Our focus is on understanding LLD within GRPO and proposing NTHR, a principled, lightweight fix that addresses this issue directly.
>
> --------  Weaknesses 4:  hyperparamters ---------
>
> **Answer**:  Thank you for the comment. To clarify, our method introduces only two hyperparameters: α and β. The threshold tau is determined by β as shown in line 259, Among table 1’s experiments, β is fixed to 1 in all experiments to influence, meaning that in practice, only one hyperparameter α is varied.
> Moreover, our contribution is not limited to improving final task performance through tuning. A central goal of the paper is to understand and address the LLD phenomenon. We conduct both theoretical analysis (e.g., Theorem 4.4) and empirical studies (e.g., Figure 1 and 3) to diagnose how and why LLD arises during training. The proposed mitigation is grounded in these insights, not in hyperparameter optimization.

---

> ### Author Response · Authors · 2025-08-04
> **Reaching the End of the Public Discussion Phase**
>
> Dear Reviewer,
>
> Thank you very much for your efforts in the reviewing process, we have added experiments and analysis to answer your questions and hope that our responses have sufficiently addressed the concerns you raised. We welcome further discussion if you have more questions and suggestions.
>
> As the discussion deadline is approaching, we would be very grateful if you could take a moment to review our reply. Thank you for your time and consideration.
>
> Best
> Authors

---

> > ### Comment · Reviewer_Nhas · 2025-08-07
> >
> > Thanks for your rebuttal. My main concerns were addressed I've raised my score.

---

> > > ### Author Response · Authors · 2025-08-07
> > > **Thank you!**
> > >
> > > Dear Reviewer Nhas:
> > >
> > > Thank you for taking the time to review our rebuttal. We sincerely appreciate your thoughtful engagement and are glad to hear that our response addressed your concerns. We truly value your updated score and are grateful for your constructive feedback throughout the review process.
> > >
> > > Best
> > >
> > > Authors

---

### Official Review · Reviewer_JEcX · 2025-07-03

**Clarity:** 4
**Significance:** 3
**Originality:** 3
**Rating:** 4
**Confidence:** 5

**Summary:**

This paper investigates the effect of negative gradients in Group Relative Policy Optimization (GRPO), a popular reinforcement learning method for fine-tuning large language models. The authors identify a phenomenon called Lazy Likelihood Displacement (LLD), where the likelihood of correct responses fails to increase—or even decreases—during training. To address this, they propose Negative Token Hidden Reward (NTHR), a token-level selective penalization strategy that reduces the impact of harmful negative gradients. Theoretical analysis and extensive experiments on math reasoning benchmarks demonstrate that NTHR effectively mitigates LLD and improves model performance across various scales.

**Questions:**

No

**Ethical Concerns:**

["NO or VERY MINOR ethics concerns only"]

**Final Justification:**

My concerns—such as the comparison with DPO-style methods, generalization of the embedding-based strategy, clarity of certain figures, and the need for stronger empirical analysis—have all been addressed well.

**Limitations:**

yes

**Quality:**

3

**Strengths And Weaknesses:**

Strengths:

This paper identifies and analyzes a previously overlooked issue—Lazy Likelihood Displacement (LLD)—in GRPO-based reinforcement learning for LLMs, and proposes a novel solution, NTHR, that selectively reduces penalties on certain tokens to mitigate this effect. The work is well-motivated, theoretically grounded, and supported by comprehensive experiments across multiple model sizes, demonstrating consistent performance gains and strong practical relevance.

Weaknesses:

1. The authors should extend and adapt the existing methods proposed to mitigate the reduced probabilities of preferred responses in DPO to the GRPO setting, and include comparisons with the proposed NTHR strategy. This would better support the related claims made in the introduction and highlight the advantages of the proposed approach.

2. Intuitively, directly applying an NLL loss to the positive samples can increase their probabilities and thereby alleviate the identified Lazy Likelihood Displacement phenomenon, as has been done in [1]. How does this approach compare to the proposed NTHR strategy in terms of advantages and disadvantages?

3. I am curious whether the NTHR strategy might lead to under-penalization of incorrect responses, potentially resulting in the retention of erroneous patterns and degradation of model behavior.

4. The proposed strategy relies on token embedding similarity to identify tokens for reduced penalization, which may be sensitive to the quality of the embedding space. Therefore, it would be beneficial for the authors to evaluate the generalization and stability of the method on a wider range of models, such as the LLaMA-3 series.

5. The authors could cite [2–3] to better support the use of final-layer embeddings.

If the authors address my concerns through additional experiments and discussion, I would be happy to raise my score to 4 or 5.

[1] VAPO: Efficient and Reliable Reinforcement Learning for Advanced Reasoning Tasks. arXiv 2025

[2] Mitigating Reward Overoptimization via Lightweight Uncertainty Estimation. NeurIPS 2024

[3] The Energy Loss Phenomenon in RLHF: A New Perspective on Mitigating Reward Hacking. ICML 2025

---

> ### Author Response · Authors · 2025-07-31
> **Weaknesses1 adapt DPO method & weakness 2 add NLL loss.**
>
> **Apply DPO methods**:  While DPO-inspired strategies are indeed useful, most of them focus on reduced likelihood rather than Lazy Likelihood Displacement (LLD), and are designed for off-policy settings—making them difficult to apply directly to GRPO. That said, we were inspired by DPO's core motivation: ensuring that the likelihood of preferred responses does not fall below that of a reference model.
> Following this idea and motivated by your suggestion, we implemented **SMAUG** [1] on Llama-1B-Ins, a simple regularization term that adds `max(0, π_ref - π_θ)` for correct responses (y > 0). As shown below, SMAUG achieves performance comparable to GRPO. Moreover, we observe that fewer than 8% of training iterations exhibit reduced likelihood for correct responses, aligning with our observations in Figure 1 that mostly are LLD. This further supports our decision to focus on LLD as the primary optimization issue.
>
> | Method         | math500 | minerva_math | olympiad | aime24 | amc23 | avg   |
> |----------------|---------|--------------|----------|--------|-------|--------|
> | GRPO     | 37.0    | 7.4          | 9.3      | 3.3    | 20.0  | 15.4  |
> | + SMAUG   | 37.6    | 9.6          | 7.9      | 3.3    | 17.5  | 15.18 |
>
> Thank you for the comment, we will add this experiment in the revision.
>
> **NLL as a complementary**: Your suggestion to incorporate an NLL loss on correct responses aligns well with the LLD issue we diagnose. Specifically, adding a supervised NLL term helps reinforce the model’s confidence on correct responses. When we added NLL loss to GRPO and train with Llama-1B-Ins, we observed improved performance across several tasks. The NLL objective addresses LLD from a complementary angle—reinforcing correct responses via positive gradients, whereas NTHR mitigates harmful negative updates. We observe both methods improve the performance.
>
> | Method         | math500 | minerva_math | olympiad | aime24 | amc23 | avg    |
> |----------------|---------|--------------|----------|--------|-------|--------|
> | GRPO         | 37.0    | 7.4          | 9.3      | 3.3    | 20.0  | 15.4   |
> | + NLL Loss     | 36.0    | **9.6**          | **11.0**     | 3.3    | **25.0**  | **16.98**  |
> | + NTHR    | **37.4**    | 8.1          | **11.0**     | 3.3    | 20.0  | 15.96  |
>
>  Thanks for your suggestion! We will add this in the revision.
>
> [1] Smaug: Fixing Failure Modes of Preference Optimisation with DPO-Positive

---

> ### Author Response · Authors · 2025-07-31
> **Weakness3&4&5**
>
> ----------Weakness3: NTHR effectiveness -----------
>
> **Answer**: Thank you for this insightful question. Our NTHR method is explicitly designed to avoid indiscriminate under-penalization. It selectively reduces the penalty only on specific negative tokens within incorrect responses that have been shown — both theoretically (Theorem 4.4) and empirically (Fig. 3) — to contribute to Lazy Likelihood Displacement by overlapping with key reasoning steps in correct responses.
> As also shown in Fig 3, the selected tokens can be logically or step-wise correct fragments embedded in otherwise incorrect answers. By attenuating the penalty on such tokens, NTHR preserves useful reasoning patterns rather than reinforcing errors. Importantly, all other negative tokens that do not align with correct responses remain fully penalized, ensuring that genuinely erroneous content is still suppressed.
> Empirically, this selective strategy consistently improves both the likelihood gain of correct responses and downstream task performance compared to baseline GRPO (see Fig. 4 and Tab. 2), indicating that NTHR mitigates LLD without degrading overall model behavior.
>
>
> -------------Weakness 4: quality of the embedding space and results using llama -------
>
> **Answer**: Thank you very much for your question! We now add results on Llma3.2-1B-Ins. As shown in the table below, applying our NTHR method to LLaMA3.2-1B-Ins yields consistent improvements over the GRPO baseline, mirroring the trends observed with Qwen2.5. This suggests that our approach generalizes well and is not overly sensitive to the specific embedding space of a given model.
>
> | Method         | math500 | minerva_math | olympiad | aime24 | amc23 | avg    |
> |----------------|---------|--------------|----------|--------|-------|--------|
> | GRPO     | 37.0    | 7.4          | 9.3      | 3.3    | 20.0  | 15.4   |
> | +NTHR     | **37.4**    | **8.1**          | **11.0**     | 3.3    | 20.0  | **15.96**  |
>
> -------------Weakness 5: related works -------------
>
> **Answer**: Thank you very much for your suggestion. We agree [2-3] are related and we will discuss them in our revised version.
>
> We hope the responses above answer your questions and will prompt you to kindly reconsider your score. Thanks again for the time and useful suggestions.
>
> [2] Mitigating Reward Overoptimization via Lightweight Uncertainty Estimation. NeurIPS 2024
>
> [3] The Energy Loss Phenomenon in RLHF: A New Perspective on Mitigating Reward Hacking. ICML 2025

---

> > ### Comment · Reviewer_JEcX · 2025-08-05
> >
> > Thank you for the detailed response. My concerns—such as the comparison with DPO-style methods, generalization of the embedding-based strategy, clarity of certain figures, and the need for stronger empirical analysis—have all been addressed well. I will raise my score.

---

> > > ### Author Response · Authors · 2025-08-06
> > > **Thank you!**
> > >
> > > Dear Reviewer JEcX:
> > >
> > > Thank you for taking the time to review our rebuttal. We sincerely appreciate your engagement and are delighted to hear that our response has addressed your concerns. Your updated score is greatly valued, and we are grateful for your constructive feedback throughout this process.
> > >
> > > Best
> > >
> > > Authors

---

> ### Author Response · Authors · 2025-08-04
> **Reaching the End of the Public Discussion Phase**
>
> Dear Reviewer,
>
> Thank you very much for your efforts in the reviewing process, we have added experiments and analysis to answer your questions and hope that our responses have sufficiently addressed the concerns you raised. We welcome further discussion if you have more questions and suggestions.
>
> As the discussion deadline is approaching, we would be very grateful if you could take a moment to review our reply. Thank you for your time and consideration.
>
> Best
> Authors

---

### Note · Authors · 2025-08-14

We sincerely thank the AC and reviewers for their time, constructive feedback, and support.

Our work addresses a critical but underexplored gap in GRPO-based reinforcement learning for reasoning models: **handling three intertwined problems** simultaneously.
- **Problem A — Newly identified Lazy Likelihood Displacement (LLD)**: We show that negative gradients on incorrect responses can inadvertently lower or barely increase the likelihood of correct responses. This phenomenon was not apparent in prior GRPO literature.
- **Problem B — Understanding and validating the existence of LLD**: Through theoretical analysis (e.g., Theorem 4.4), we reveal that correct responses can suffer from LLD when semantically similar but incorrect alternatives are present.
- **Problem C — Practical mitigation without losing GRPO’s advantages**: We propose a solution that addresses Problems A and B while preserving GRPO’s online training loop and maintaining interpretability.

**Our Unified Solution** :
- **Diagnose & explain LLD (A)**: We provide both empirical evidence and a mechanistic explanation showing negative gradients suppress correct responses‘ likelihood.
- **Group-aware token influence analysis (B)**: Using correct responses as anchors, we identify which tokens in negative responses most strongly depress the likelihood of the correct responses.
- **Negative Token Hidden Reward (NTHR) (B→C)**: A selective, token-level adjustment that downweights the most “harmful” tokens, thereby preserving beneficial overlap and retaining GRPO’s online, data-efficient setup.
- **Interpretability**: The identified harmful negative gradients align with key reasoning steps, enabling transparent analysis and model debugging.

**Strength in Practice**:   Across math-reasoning benchmarks and Qwen models ranging from 0.5B to 3B parameters, GRPO+NTHR consistently outperforms standard GRPO. To further support the importance and generality of addressing LLD, we added results for SMAUG (DPO-positive) and +NLL baselines (validates the harmful impact of LLD), as well as evidence of LLD in MedCQA and code generation tasks. We also included results on LLaMA, demonstrating that our method generalizes across model families.

**Takeaway**: Our work (1) identifies and validates the LLD phenomenon, (2) uncovers its underlying mechanism, and (3) proposes NTHR to address it and enhance performance. These contributions offer new insights on how negative gradients influence RL training dynamics in LLMs.

---

### Decision · Program_Chairs · 2025-09-17

**Decision:**

Accept (poster)

**Comment:**

This paper identifies Lazy Likelihood Displacement (LLD) phenomena in Group Relative Policy Optimization (GRPO), where the likelihood of correct answers stagnates. The authors trace this to uniform penalties on incorrect responses and propose Negative Token Hidden Reward (NTHR) to selectively penalize only the tokens causing LLD. Experiments on 0.5B, 1.5B, and 3B models show that NTHR improves performance.

The main point of contention was the paper's lack of theoretical grounding. Reviewers noted the analysis lacked the explanatory clarity of traditional RL theory. The authors defended their work as a behavioral analysis, which is a fair perspective. However, a key finding complicates their narrative, a simpler variant (THR) performs even better, suggesting the paper's focus on negative token penalization is an incomplete explanation.

Given that a complete theoretical treatment of modern LLMs is often intractable, the valuable empirical insights this paper provides into a real training problem are sufficient. It sheds light on an interesting phenomenon even if it doesn't provide all the answers. Therefore, I recommend acceptance.